# Shapley Neuron Values for Continual Learning: Which Neurons Matter Most?

**Mohammad Ali Vahedifar** [1]   **Abhisek Ray** [1]   **Qi Zhang** [1]

GitHub Code

## Abstract

Continual learning enables neural networks to learn tasks sequentially without forgetting previously acquired knowledge. However, neural networks suffer from catastrophic forgetting, where learning new tasks degrades performance on earlier ones. We address this problem with **Shapley Neuron Valuation (SNV)**, a principled framework that quantifies Neuron importance in continual learning, grounded in cooperative game theory. SNV selectively freezes important Neurons while keeping others plastic, enabling **buffer-free** continual learning without expanding architecture. Experiments on ImageNet-1k show that SNV consistently outperforms existing buffer-free methods. In particular, SNV improves accuracy by **+2.88%** in the class incremental learning and **+6.46%** in the task incremental learning scenarios compared to the second baseline.

## 1. Introduction

Sequential training of neural networks often leads to catastrophic forgetting, where learning new information causes the model to lose previously acquired knowledge. This challenge is especially pronounced when models are exposed to data over time rather than all at once (Vahedifar et al., 2026). Continual learning addresses this issue by providing a framework that learns sequential data across different tasks while preserving prior knowledge. As a result, models can adapt to new tasks without compromising their performance on earlier ones (Wang et al., 2024).

Prior research on continual learning primarily falls into three categories: *regularization-based, memory-based*, and *dynamic architecture* approaches. (i) Regularization-based methods, such as Elastic Weight Consolidation (EWC) (Kirkpatrick et al., 2017), Synaptic Intelligence (SI) (Zenke et al., 2017), and Learning without Forgetting (LwF) (Li & Hoiem, 2018) constrain parameter updates to reduce interference with previously learned tasks. (ii) Memory-based approaches, including Incremental Classifier and Representation Learning (iCaRL) (Rebuffi et al., 2017), Dark Experience Replay (DER++) (Buzzega et al., 2020), Pooled Outputs Distillation Network (PODNet) (Douillard et al., 2020), CO-transport for class Incremental Learning (COIL) (Zhou et al., 2021), and Gradient Episodic Memory (GEM) (Lopez-Paz & Ranzato, 2017), mitigate forgetting by storing and replaying examples or their representations from past tasks. (iii) Dynamic architecture methods, such as Progressive Neural Networks (PNN) (Rusu et al., 2022), DYnamic TOken eXpansion (DyTox) (Douillard et al., 2022), and Memory-efficient Expandable MOdel (MEMO) (Zhou et al., 2023), expand the model over time to preserve task-specific representations.

Despite their success, memory-based methods violate the strict continual learning setting, where the current task's data can only be accessible, as in (Rebuffi et al., 2017; Buzzega et al., 2020; Douillard et al., 2022; Zhou et al., 2023). As a result, it raises concerns about data privacy and the growing memory storage cost. On the other hand, dynamic architecture methods expand the model as tasks accumulate, leading to unbounded parameter and memory growth. Such expansion becomes computationally impractical for large-scale settings or long task sequences, as in (Buzzega et al., 2020; Rusu et al., 2022). Even regularization-based methods, although buffer-free, often require additional task-specific modules (e.g., task classifiers, bias correction branches), auxiliary losses, or architectural constraints. Their performance deteriorates when tasks become highly heterogeneous, as in (Kirkpatrick et al., 2017; Vahedifar & Zhang, 2026). The critical question is: *How can we achieve forget-resistant continual learning method within the fixed capacity of the backbone network without revisiting past data, and without growing the architecture?*

This work is motivated by the observation that modern neural networks are highly over-parameterized, containing large reservoirs of redundant capacity (Frankle et al., 2020). Rather than adding new parameters or storing past samples, we argue that a fixed-capacity model already contains

---

[1]DIGIT and Department of Electrical and Computer Engineering, Aarhus University, Denmark. Correspondence to: Mohammad Ali Vahedifar <av@ece.au.dk>, Qi Zhang <qz@ece.au.dk>.

*Proceedings of the $43^{rd}$ International Conference on Machine Learning*, Seoul, South Korea. PMLR 306, 2026. Copyright 2026 by the author(s).

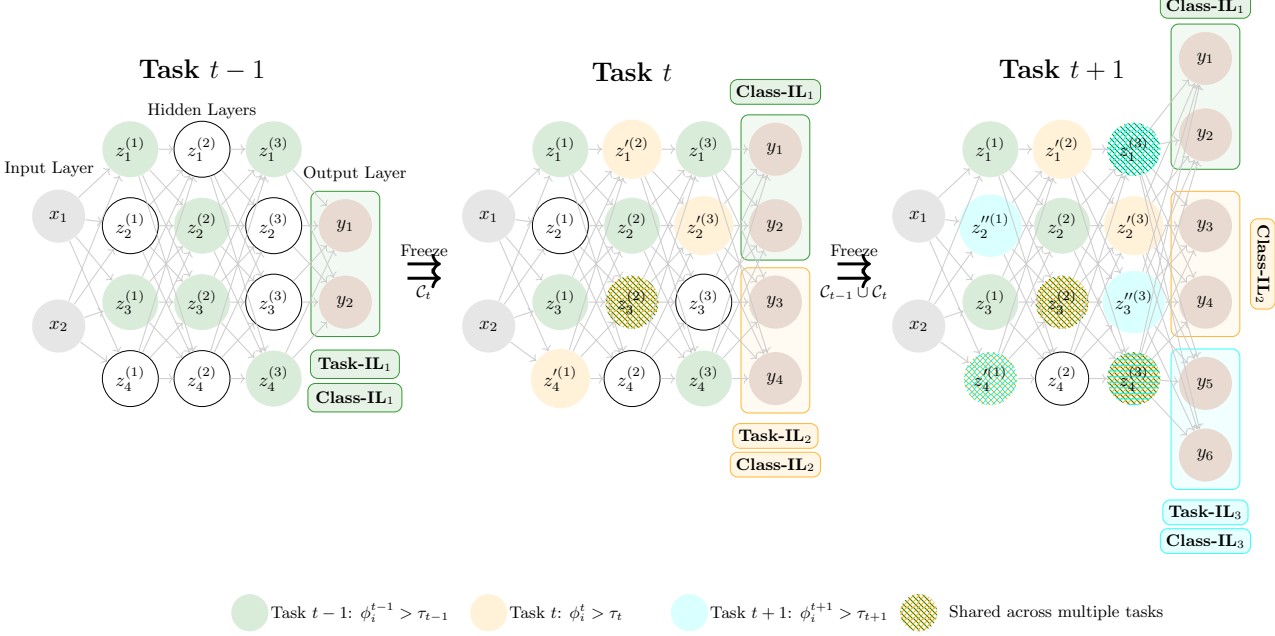

*Figure 1.* An illustration of Shapley Neuron Values where for each task task $t-1$, task $t$, and task $t+1$, we identify and freeze the Neurons whose Shapley Values fall within the top $r\%$. Neurons marked with specific colors indicate that the same Neuron appears within the top $r\%$ for $t$ specific tasks. $\tau_i$ is each task's top $r\%$ threshold.

sufficient internal degrees of freedom to encode long task sequences, provided that the most important Neurons for each task can be reliably identified and structurally preserved. We propose **Shapley Neuron Values (SNV)**, a buffer-free, without architecture expansion continual learning framework that leverages the inherent over-parameterization of neural networks to identify expert subnetworks for each task. Our key insight is to identify and freeze the most influential Neurons for each task using a principled valuation mechanism based on Shapley Values from cooperative Game Theory (GT). This ensures a clear separation between: **Stable Neurons** that encode knowledge from previous tasks. and **Plastic Neurons** that remain free to learn new tasks.

**Related work.** Due to space constraints, we defer our review of prior works to Appendix B. There, we provide an extended discussion of continual learning approaches.

## 2. Shapley Neuron Values

The exploration of sparse subnetworks offers a compelling paradigm for addressing catastrophic forgetting within a continual learning framework. The methodological core involves systematically ascertaining the most salient Neurons essential for the current task and freezing their corresponding weights, as shown in Fig. 1. This preservation mechanism effectively stabilizes the knowledge pertinent to previously learned tasks, ensuring requisite stability. Simultaneously, the remaining pool of Neurons' capacity retains

plasticity, allowing for the flexible representation of subsequent tasks. This stability–plasticity trade-off disentangles knowledge retention from new task learning by reducing cross-task interference. By maintaining past information stability while allowing for flexible adaptation to new tasks, the system supports both scalable and sequential learning.

**Problem Statement.** Consider a supervised learning setup where, $\mathcal{T} = \{T_t\}_{t=1}^T$ tasks arrive sequentially corresponding to $\mathcal{D} = \{\mathcal{D}_t\}_{t=1}^T$ data. Each task $t$ has dataset $\mathcal{D}_t = \{(x_j^t, y_j^t)\}_{j=1}^{k_t}$ with $k_t$ samples. In continual learning, when learning task $t$, only $\mathcal{D}_t$ is accessible. Since modern architectures (e.g., ResNet) are predominantly convolutional, we define the fundamental unit of importance hereafter referred to as a "Neuron" as a convolutional filter (or kernel). Let the network $\mathcal{N}$ consist of $L$ layers, where the $l$-th layer contains $C_l$ filters. The total set of Neurons is $\mathcal{M} = \{m_i\}_{i=1}^N$, where $N = \sum_{l=1}^L C_l$ is the total number of filters in the network. In our setting, $V$ is a black box that takes any network as input and returns a score, such as accuracy, loss, or disparity. The performance on the full model is denoted as $V(\mathcal{M})$. Mathematically, we want to partition the overall performance metric $V(\mathcal{M})$ among model Neurons such that $\sum_{i=1}^N \phi_i(V, \mathcal{M}) = V(\mathcal{M})$, where $\phi_i$ is Neuron values.

To assess the contribution of individual Neurons in the model, we mask selected components by replacing a filter's output with its mean response over a set of validation data. This procedure blocks the flow of information through

that filter while preserving the average statistics of the signal passed to subsequent layers, which would not be the case if the output were replaced with zeros. We consider subsets $\mathcal{S} \subseteq \mathcal{M}$ representing subnetworks, and write $V(\mathcal{S})$ to denote the model's performance after all Neurons in $\mathcal{M} \setminus \mathcal{S}$ have been replaced by their mean activations. Note that $V(\emptyset)$ corresponds to all Neurons replaced by their mean responses, yielding a well-defined baseline. The model is not retrained after this modification; all parameters remain fixed, and we directly evaluate the test performance $V(\mathcal{S})$. This choice avoids the computational cost that would arise from fine-tuning the network for every possible subset $\mathcal{S}$.

Given network $\mathcal{N}$, we seek a Neuron importance-based mask $S_t \in \{0,1\}^N$ for task $T_t$. The masked parameters are obtained via broadcasting: $\mathcal{M} \odot S_t = \{w_1.S_t^{(1)}, ..., w_N.S_t^{(N)}\}$. We introduce a sparsity ratio $c \in (0,1)$ (e.g., $c = 1/T$) to limit the network capacity allocated to each task. The optimization problem is:

$$S_t^* = \arg \min_{S_t \in \{0,1\}^N} \frac{1}{k_t} \sum_{j=1}^{k_t} \mathcal{L}(f(x_j^t; \mathcal{M} \odot S_t), y_j^t), \quad (1)$$

$$s.t \quad \|S_t\|_0 \leq \lfloor c \cdot N \rfloor.$$

Here, $\|S_t\|_0$ denotes the $L_0$ norm (count of non-zero elements) of the mask, and $N$ is the total number of filters in the network. The constraint ensures that the number of active Neurons does not exceed the budget defined by $c$. Solving this combinatorial optimization exactly is intractable. Instead, we compute the Shapley Neuron Value $\phi_i$ for each Neuron and construct the mask by selecting the $\lfloor c \cdot N \rfloor$ Neurons ranked by $\phi_i$, i.e., $S_t^*(i) = 1$ if and only if $\phi_i$ is among the $\lfloor c \cdot N \rfloor$ largest values. Thus, a subnetwork for each task is obtained by $\mathcal{M}^* = \mathcal{M} \odot S_t^*$.

### 2.1. Neuron Valuation Framework

Our goal is to compute a Neuron value $\phi_i(V, \mathcal{M}) \in \mathbb{R}$ to quantify the importance of the $i$-th Neuron. For simplicity, we use $\phi_i$ to denote $\phi_i(V, \mathcal{M})$. We define that $\phi_i$ should satisfy the following properties:

*Axiom* 1 (Efficiency). The total value is distributed among all Neurons:

$$\sum_{i \in \mathcal{M}} \phi_i = V(\mathcal{M}). \quad (2)$$

*Axiom* 2 (Null Contribution). If adding a specific Neuron does not improve performance, no matter which subset it is added to, then it should have zero value:

$$\forall \mathcal{S} \subseteq \mathcal{M} \setminus \{i\}, V(\mathcal{S}) = V(\mathcal{S} \cup \{i\}) \Rightarrow \phi_i = 0. \quad (3)$$

Note, $\mathcal{M} \setminus \{i\}$ means a set of Neurons without Neuron $m_i$. This applies to the rest of the equations.

*Axiom* 3 (Symmetry). If two Neurons contribute equally to the model, then their values must be the same. For Neurons $i$ and $j$ and any $\mathcal{S} \subseteq \mathcal{M} \setminus \{i, j\}$:

$$V(\mathcal{S} \cup \{i\}) = V(\mathcal{S} \cup \{j\}) \Rightarrow \phi_i = \phi_j. \quad (4)$$

*Axiom* 4 (Linearity). For a given Neuron, its effect on the sum of two functions equals the sum of its individual effects on each function.

$$\phi_i(V_1 + V_2) = \phi_i(V_1) + \phi_i(V_2). \quad (5)$$

The contribution formula that uniquely satisfies all these axioms is defined in the following Theorem.

**Theorem 2.1** (Shapley Neuron Value). *Any Neuron valuation $\phi_i(V, \mathcal{M})$ satisfying the above-mentioned Axioms must have the form:*

$$\phi_i = \sum_{\mathcal{S} \subseteq \mathcal{M} \setminus \{i\}} \frac{|\mathcal{S}|!(|\mathcal{M}| - |\mathcal{S}| - 1)!}{|\mathcal{M}|!} [V(\mathcal{S} \cup \{i\}) - V(\mathcal{S})],$$
$$(6)$$

*where $\phi_i$ is called the "Shapley Neuron Value" of Neuron $m_i$.*

*Proof.* The expression in Eq. 6 is identical to the Shapley value defined in GT (Shapley, 1953), which motivates the term Shapley Neuron Value. This follows from the fact that the Shapley Value is the unique solution satisfying a specific set of axioms, and the Neuron valuation problem can be cast into the same mathematical framework. In this view, Neuron valuation is a cooperative game in which Neurons act as players and model performance serves as the payoff. Different subsets of Neurons yield different prediction performance, analogous to how different coalitions of players achieve varying rewards. The Shapley Neuron Value then assigns each Neuron a fair contribution, following the same axiomatic principles that govern the original Shapley Value. □

Importantly, unlike binary importance methods that assign $\{0, 1\}$ scores to Neurons (e.g., WSN), the Shapley Neuron Values $\phi_i \in \mathbb{R}$ provide a continuous importance ranking. This continuous valuation enables principled Neuron selection under tight capacity budgets $c$, where the relative ordering among candidate Neurons becomes critical. The final task mask $S_t \in \{0, 1\}^N$ is then derived by thresholding this continuous ranking at the budget $c$.

### 2.2. Cumulative Shapley Mask

To prevent catastrophic forgetting, we selectively update model parameters by allowing gradients to flow only through Neurons that have not been allocated to previous tasks. Neurons identified as *important* for earlier tasks are

frozen to preserve the acquired knowledge. We define an accumulated binary mask at the Neuron level for task $t$:

$$\mathbf{B}_{t-1} = \bigcup_{i=1}^{t-1} S_i, \tag{7}$$

where $\mathbf{B}_{t-1} \in \{0,1\}^N$ marks the set of all filters used in tasks 1 through $t-1$. To apply this at the parameter level, let $\theta$ denote the set of all trainable weights in the network. We define a parameter-level freezing mask $\mathbf{M}_{t-1}$ of the same dimension as $\theta$. For each weight $\theta_j$, the mask is defined as:

$$(\mathbf{M}_{t-1})_j = \begin{cases} 0 & \text{if weight } \theta_j \in m_i \text{ where } (\mathbf{B}_{t-1})_i = 1 \\ 1 & \text{otherwise.} \end{cases} \tag{8}$$

For an optimizer with learning rate $\eta$, the parameter update rule for task $t$ is:

$$\theta \leftarrow \theta - \eta \left( \frac{\partial \mathcal{L}}{\partial \theta} \odot \mathbf{M}_{t-1} \right). \tag{9}$$

This formulation ensures that the gradient $\frac{\partial \mathcal{L}}{\partial \theta_j}$ is zeroed out for any weight belonging to a frozen subnetwork.

### 2.3. Estimating Shapley Neuron Values

Although Neuron Shapley has desirable theoretical properties, its exact computation is prohibitively expensive. Because the number of possible subsets $\mathcal{S}$ grows exponentially, computing the Shapley Value for a single Neuron requires an exponential number of operations. Inspired by (Ghorbani & Zou, 2020), we approximate Shapley Values through:

**i. Monte Carlo Estimation:** For a model with $|\mathcal{M}|$ elements, the Shapley value of the $i$-th component can be written as:

$$\phi_i = \mathbb{E}_{\pi \sim \Pi} \left[ V(\mathcal{S}_i^\pi \cup \{i\}) - V(\mathcal{S}_i^\pi) \right], \tag{10}$$

where $\Pi$ denotes the uniform distribution over all $|\mathcal{M}|!$ permutations of the model elements, and $\mathcal{S}_i^\pi$ is the set of elements appearing before element $i$ in permutation $\pi$ (or the empty set if $i$ is first). Approximating $\phi_i$ thus reduces to estimating the mean of a random variable. For a chosen error bound, Monte Carlo estimation provides an unbiased estimator of $\phi_i$.

**ii. Truncation:** The main computational cost in the expression above comes from evaluating the marginal contribution:

$$V(\mathcal{S}_i^\pi \cup \{i\}) - V(\mathcal{S}_i^\pi). \tag{11}$$

When $\mathcal{S}_i^\pi$ is small, the model's performance $V(\mathcal{S}_i^\pi)$ degrades toward zero due to the removal of many network filters. Therefore, we skip marginal computations for early elements in a sampled permutation $\pi$ when $\mathcal{S}_i^\pi$ is below

a predefined performance threshold, treating the model as effectively non-functional. This truncation yields significant computational savings, often close to an order of magnitude.

**iii. Multi-Armed Bandit:** The objective is to reliably identify the top-$k$ Neurons. Instead of computing the marginal contribution for every Neuron at each iteration, we restrict sampling to those Neurons whose current confidence intervals still overlap with the top-$k$ largest estimated value. In other words, we sample only the Neurons whose lower and upper bounds straddle the current top-$k$ position. When no Neurons meet this condition, the algorithm concludes that the top-$k$ Neurons are confidently distinguished from the rest within the specified error tolerance. Our goal aligns with multi-armed bandit (MAB) settings (Jamieson & Talwalkar, 2016; Li et al., 2017). An MAB is a decision-making problem where you have several arms (choices), and each choice gives a reward, but you don't know which one is best. The MAB framework supports this goal through two complementary steps. **(a) Exploitation:** The algorithm allocates more samples to Neurons that currently show high estimated Shapley Values, allowing it to refine estimates for the most promising candidates. **(b) Exploration:** It also samples Neurons with limited data or high uncertainty, reducing the risk of overlooking a genuinely important Neuron due to a few early low-value observations. Algorithm 1 summarizes the steps of the SNV method.

**Notation for Algorithm.** Let $a_i(x)$ denote the activation of Neuron $i$ for input $x$, and let $\mu_i$ be its mean activation over the current task's data. The truncation threshold $\tau$ defines a minimum performance floor: when $V(\mathcal{S}) \leq \tau$, marginal contributions are skipped for efficiency. Finally, $z_\alpha$ is the standard-normal critical value at confidence level $\alpha$.

## 3. Experiments

**Scenarios:** We conducted a series of experiments to evaluate the performance in Class Incremental Learning (CIL) and Task Incremental Learning (TIL) scenarios (See section D in the Appendix).

**Evaluation Metrics:** To assess the ability of each method to perform effective CL and battle catastrophic forgetting, we use Average Accuracy (ACC), Backward Transfer (BWT), and Plasticity-Stability (PS) (See section E in the Appendix).

**Datasets:** We conduct experiments on CIFAR-100 (Krizhevsky & Hinton, 2009), Tiny-ImageNet (Le & Yang, 2015), and ImageNet-1k (Deng et al., 2009).

**Methods:** Our primary focus is to compare our approach with buffer-free methods, but we also incorporate memory-based and architecture-based methods. We used Stochastic Gradient Descent (SGD) as a lower bound and joint training as an upper bound to establish performance bounds. Addi-

---

**Algorithm 1** Continual Learning with Shapley Neuron Valuation (SNV)

**procedure** TRAIN($f_\theta$, $\{\mathcal{D}_t\}_{t=1}^T$, $c$, $\tau$, $\alpha$)
  Initialize $\theta$ via He initialization
  $\mathbf{B}_0 \leftarrow \mathbf{0}^N$
  $R \leftarrow 0 \in \mathbb{R}^{T \times T}$
  **for** $t = 1, \ldots, T$ **do**
    Construct freezing mask:
$$(\mathbf{M}_{t-1})_j \leftarrow \begin{cases} 0 & \text{if } \theta_j \in m_i, \ (\mathbf{B}_{t-1})_i = 1 \\ 1 & \text{otherwise} \end{cases}$$
    **for** each epoch **do**
      **for** $(x, y) \in \mathcal{D}_t^{\text{train}}$ **do**
        $\mathcal{L} \leftarrow \text{Loss}(f_\theta(x, t), \ y)$
        $\theta \leftarrow \theta - \eta\,(\nabla_\theta \mathcal{L} \odot \mathbf{M}_{t-1})$
      **end for**
    **end for**
    $S_t \leftarrow \text{ESTIMATESNV}(f_\theta, \mathcal{D}_t^{\text{val}}, c, \tau, \alpha)$
    $\mathbf{B}_t \leftarrow \mathbf{B}_{t-1} \cup S_t$
    Store task head $h_t$
    $R_{t,:} \leftarrow \text{EVALUATE}(f_\theta, \{\mathcal{D}_k^{\text{test}}\}_{k=1}^t)$
  **end for**
  **return** $f_\theta$, $R$, $\{S_t\}_{t=1}^T$, $\{h_t\}_{t=1}^T$
**end procedure**

---

**procedure** ESTIMATESNV($f_\theta$, $\mathcal{D}_{\text{val}}$, $c$, $\tau$, $\alpha$)
  **for** $i = 1, \ldots, N$ **do**
    $\mu_i \leftarrow \frac{1}{|\mathcal{D}_{\text{val}}|} \sum_{x \in \mathcal{D}_{\text{val}}} a_i(x)$
  **end for**
  $\hat{\phi}_i \leftarrow 0$, $n_i \leftarrow 0$, $\sigma_i^2 \leftarrow 0 \ \ \forall i$
  $k \leftarrow \lfloor c \cdot N \rfloor$, $\mathcal{A} \leftarrow \mathcal{M}$

**while** $\mathcal{A} \neq \emptyset$ **do**
  Sample $\pi \sim \text{Uniform}(\Pi)$
  $\mathcal{S} \leftarrow \emptyset$
  **for** $j = 1, \ldots, N$ **do**
    $i \leftarrow \pi(j)$
    **if** $i \in \mathcal{A}$ and $V(\mathcal{S}) > \tau$ **then**
      $\Delta_i \leftarrow V(\mathcal{S} \cup \{i\}) - V(\mathcal{S})$
      Update $\hat{\phi}_i, \sigma_i^2, n_i$ with $\Delta_i$
    **end if**
    $\mathcal{S} \leftarrow \mathcal{S} \cup \{i\}$
  **end for**
  $\delta_i \leftarrow z_\alpha \cdot \sigma_i / \sqrt{n_i} \ \ \forall i : n_i > 0$
  $\phi^{(k)} \leftarrow k\text{-th largest in } \{\hat{\phi}_i\}$
  $\mathcal{A} \leftarrow \{i : |\hat{\phi}_i - \phi^{(k)}| < \delta_i\}$
**end while**
$S_i \leftarrow \mathbf{1}[i \in \arg\text{top-}k(\{\hat{\phi}_i\})] \ \forall i$
**return** $S_i$
**end procedure**

---

**procedure** EVALUATE($f_\theta$, $\{\mathcal{D}_k^{\text{test}}\}_{k=1}^t$)
  $r \leftarrow 0 \in \mathbb{R}^t$
  **for** $k = 1, \ldots, t$ **do**
    $r_k \leftarrow 0$
    **for** $(x, y) \in \mathcal{D}_k^{\text{test}}$ **do**
      $r_k \leftarrow r_k + \text{accuracy}(f_\theta(x, k), \ y)$
    **end for**
    $r_k \leftarrow r_k / |\mathcal{D}_k^{\text{test}}|$
  **end for**
  **return** $r$
**end procedure**

---

tionally, we compare SNV against representative buffer-free continual learning methods, such as EWC (Kirkpatrick et al., 2017), SI (Zenke et al., 2017), LwF (Li & Hoiem, 2018), Prediction Error-based Classification (PEC) (Zajac et al., 2024), Winning SubNetwork (WSN) (Kang et al., 2022), SpaceNet (Sokar et al., 2021), Neuro-Inspired Stability-Plasticity Adaptation (NISPA) (Gurbuz & Dovrolis, 2022), Discriminative and Consistent Network (DCNet) (Wang et al., 2025b), and No Forgetting Learning (NFL) (Vahedifar & Zhang, 2026). *Memory-based methods:* iCaRL (Rebuffi et al., 2017), DER++ (Buzzega et al., 2020), and DyTox (Douillard et al., 2022).

**Experiment Setup:** We adopt the ResNet-18 architecture (He et al., 2016) with He initialization (He et al., 2015) using the Adam optimizer (Kingma & Ba, 2015) for training. All training is averaged over 10 runs, with training using 70% of the data and testing using 20%, and 10% of the data from each task is reserved as a validation set. For experiments on CIFAR-100 and Tiny-ImageNet, we train for a maximum of 200 epochs with early stopping based on validation loss and a batch size of 64. For ImageNet-1k, we train for a maximum of 100 epochs per task with early stopping and a batch size of 128. We adopt a tuning protocol aligned with the principles of the Generalizable Two-phase Evaluation Protocol (GTEP) (Cha & Cho, 2025). Hyperparameters are selected using only the first task's validation split via grid search over the ranges specified in Table 7. Once selected, these hyperparameters are *fixed for all subsequent tasks* and are never re-tuned on later data. For memory-based methods in the CIL setting, we follow prior work (Zhou et al., 2024b) and use 2,000 exemplars for CIFAR-100 and Tiny-ImageNet, and 20,000 exemplars for ImageNet-1k. In the TIL setting, we follow (Buzzega et al., 2020) and allocate 200 exemplars. All experiments were conducted on a single NVIDIA A6000 GPU.

### 3.1. Comparison based on the CIL Scenario

Table 1 presents results on ImageNet-1k, the largest dataset in our evaluation. In the CIL scenario, SNV achieves 41.30% on 10 tasks, 34.20% on 20 tasks, and 25.60% on 50

*Table 1.* Performance comparison on ImageNet-1k across 10, 20, and 50 tasks (ResNet-18) for both CIL and TIL scenarios. Best results are highlighted for buffer-free and memory-based.

| | CIL | | | | | | | | | TIL | | | | | | | | |
|---|---|---|---|---|---|---|---|---|---|---|---|---|---|---|---|---|---|---|
| | 10 Tasks | | | 20 Tasks | | | 50 Tasks | | | 10 Tasks | | | 20 Tasks | | | 50 Tasks | | |
| Method | ACC↑ | BWT↑ | PS↑ | ACC↑ | BWT↑ | PS↑ | ACC↑ | BWT↑ | PS↑ | ACC↑ | BWT↑ | PS↑ | ACC↑ | BWT↑ | PS↑ | ACC↑ | BWT↑ | PS↑ |
| *Memory-free methods* | | | | | | | | | | | | | | | | | | |
| SNV (Ours) | **41.30** ±1.25 | −0.05 | 0.65 | **34.20** ±1.48 | −0.05 | 0.58 | **25.60** ±1.62 | −0.06 | 0.49 | **57.82** ±1.38 | 0.0 | 0.58 | **50.45** ±1.55 | 0.0 | 0.52 | **40.18** ±1.72 | 0.0 | 0.44 |
| NFL+ | 38.42 ±3.85 | −37.78 | **0.67** | 31.50 ±3.62 | −41.20 | **0.60** | 22.40 ±3.48 | −45.80 | **0.51** | 51.36 ±4.52 | −5.19 | **0.63** | 45.80 ±4.38 | −7.85 | **0.56** | 37.20 ±4.15 | −12.40 | **0.48** |
| DCNet | 37.80 ±3.72 | −38.96 | 0.66 | 30.10 ±3.40 | −42.50 | 0.58 | 20.80 ±3.25 | −47.30 | 0.48 | 50.15 ±4.30 | −6.42 | 0.60 | 44.30 ±4.12 | −9.50 | 0.53 | 35.80 ±3.90 | −14.80 | 0.44 |
| WSN | — | — | — | — | — | — | — | — | — | 48.73 ±1.85 | 0.0 | 0.52 | 43.20 ±1.72 | 0.0 | 0.46 | 35.40 ±1.58 | 0.0 | 0.38 |
| NISPA | 29.40 ±4.15 | −43.08 | 0.61 | 22.30 ±3.88 | −46.90 | 0.52 | 14.50 ±3.60 | −51.40 | 0.41 | 46.80 ±3.95 | −8.75 | 0.55 | 40.60 ±3.72 | −12.30 | 0.47 | 32.50 ±3.48 | −18.60 | 0.38 |
| NFL | 27.15 ±4.92 | −44.80 | 0.59 | 21.40 ±4.55 | −48.50 | 0.50 | 14.80 ±4.12 | −53.10 | 0.39 | 40.18 ±5.32 | −23.62 | 0.49 | 34.50 ±5.10 | −30.15 | 0.42 | 26.80 ±4.78 | −38.70 | 0.34 |
| PEC | 14.83 ±4.25 | −50.20 | 0.53 | 11.20 ±3.95 | −54.60 | 0.45 | 7.40 ±3.50 | −59.20 | 0.35 | — | — | — | — | — | — | — | — | — |
| SpaceNet | 13.50 ±4.08 | −44.28 | 0.53 | 10.40 ±3.76 | −48.50 | 0.44 | 8.30 ±3.42 | −53.80 | 0.36 | 37.20 ±4.15 | −18.40 | 0.40 | 30.80 ±3.90 | −24.60 | 0.33 | 23.10 ±3.65 | −32.40 | 0.25 |
| LwF | 11.24 ±4.38 | −47.50 | 0.51 | 8.45 ±3.90 | −52.10 | 0.41 | 5.30 ±3.15 | −57.40 | 0.30 | 42.56 ±5.12 | −58.59 | 0.38 | 35.20 ±4.85 | −65.30 | 0.31 | 25.80 ±4.50 | −74.50 | 0.22 |
| SI | 9.68 ±5.45 | −45.51 | 0.50 | 7.15 ±4.80 | −49.80 | 0.40 | 4.50 ±3.85 | −55.20 | 0.28 | 39.82 ±4.20 | −63.76 | 0.41 | 33.40 ±4.05 | −70.20 | 0.34 | 24.60 ±3.80 | −78.90 | 0.25 |
| EWC | 7.53 ±4.68 | −41.67 | 0.48 | 5.60 ±4.20 | −46.20 | 0.38 | 3.20 ±3.55 | −52.10 | 0.25 | 36.47 ±4.02 | −54.13 | 0.37 | 30.20 ±3.88 | −61.50 | 0.30 | 22.10 ±3.62 | −70.80 | 0.22 |
| *Memory-based methods* | | | | | | | | | | | | | | | | | | |
| DyTox | 40.15 ±4.30 | −35.20 | 0.58 | 33.20 ±4.12 | −38.80 | 0.50 | 24.50 ±3.90 | −43.50 | 0.42 | 59.40 ±4.65 | −6.15 | 0.58 | 52.10 ±4.48 | −8.90 | 0.50 | 42.80 ±4.20 | −13.70 | 0.42 |
| DER++ | 14.20 ±5.15 | −32.80 | 0.36 | 10.35 ±4.80 | −36.50 | 0.30 | 6.10 ±4.25 | −41.80 | 0.23 | 55.80 ±4.72 | −6.30 | 0.36 | 48.20 ±4.50 | −9.15 | 0.30 | 39.40 ±4.18 | −14.50 | 0.23 |
| iCaRL | 12.45 ±5.70 | −56.40 | 0.37 | 8.80 ±5.25 | −60.20 | 0.31 | 5.30 ±4.65 | −65.50 | 0.24 | 55.10 ±4.35 | −9.80 | 0.37 | 47.50 ±4.15 | −13.60 | 0.31 | 38.20 ±3.90 | −19.40 | 0.24 |

*Figure 2.* ACC evaluation for CIL across 10 tasks on each dataset. Each point represents the average classification accuracy evaluated after learning a given task. For example, the value at task 5 corresponds to the average accuracy of the model on the test sets of tasks 1 through 5 after completing training on task 5. For details, see Fig. 8. Solid lines: buffer-free; dashed lines: memory-based.

tasks, consistently outperforming all buffer-free baselines. Notably, SNV even surpasses DyTox (40.15%, 33.20%, 24.50%), a memory-based method that stores 20000 exemplars, demonstrating that principled Neuron selection can compensate for the absence of a replay buffer. NFL+, the second-best buffer-free method, reaches 38.42% on 10 tasks by progressively freezing parameters, but lacks a mechanism to determine *which* parameters matter most, a distinction that becomes increasingly consequential as the task sequence grows. This gap widens from 2.88% points at 10 tasks to 3.20% points at 50 tasks.

In the TIL scenario, SNV maintains zero backward transfer across all splits while achieving the highest accuracy among buffer-free methods (57.82%, 50.45%, 40.18%). Regularization-based methods degrade sharply at this scale:

EWC drops to 7.53% (CIL, 10 tasks) and SI to 4.50% (CIL, 50 tasks), as their diagonal approximations fail to capture the complex parameter interactions present in a 1000-class problem. Performance comparison on CIFAR-100 and Tiny-ImageNet across 10 and 20 tasks for the CIL scenario is summarized in Table 4 in the Appendix.

In Fig. 2, the evolution of the precision per-task further confirms that SNV maintains a consistent advantage over buffer-free baselines across datasets. Another notable observation is that large-scale datasets, such as ImageNet-1k, tend to favor memory-based methods, whereas smaller datasets, such as CIFAR-100, do not. This is consistent with theory: memory buffers help most when a small number of stored samples can capture essential information from large, diverse datasets. However, as argued in (Vahedifar & Zhang,

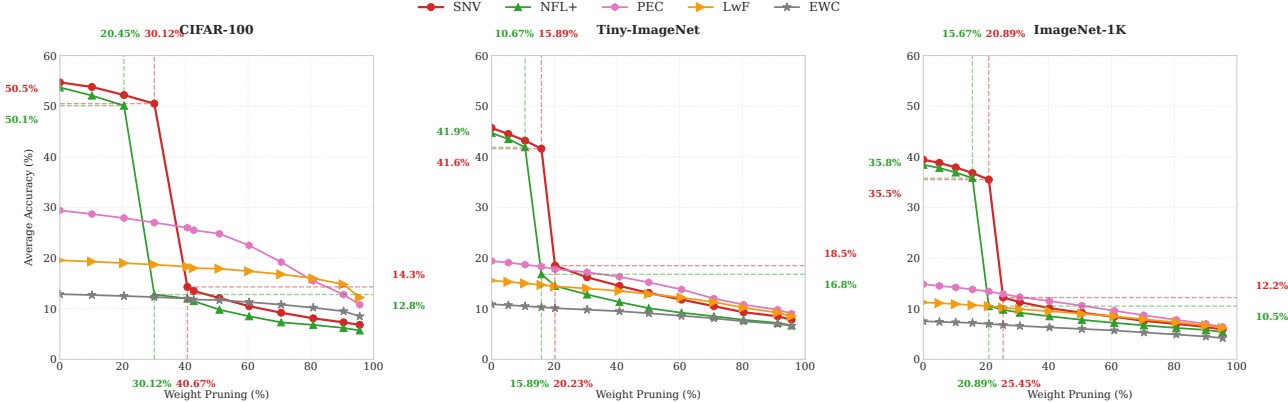

*Figure 3.* Network parameter usage efficiency across datasets in the CIL scenario for 10 tasks. The dashed vertical lines indicate the critical pruning percentage where each method experiences significant accuracy degradation. A sharper decline at lower pruning percentages indicates more efficient usage of network capacity, as it suggests the method utilizes essential parameters with minimal redundancy.

2026; Zhou et al., 2024b), many memory-based methods align more closely with sequential learning using limited replay than with strict continual learning. Comparing buffer-free methods to memory-based ones is not always fair, and using a fixed buffer size across methods can also produce misleading conclusions.

A principled mechanism for choosing both the number and the composition of stored samples (samples that contribute most to the loss or degrade performance the most) is necessary for fair evaluation. In scenarios where computational storage is not a constraint (which is uncommon in real-world systems), memory-based methods may be a reasonable approach. However, in realistic settings where storing data is costly or mandated by regulations like GDPR (Voigt & Von dem Bussche, 2017), buffer-free methods become more valuable and inevitable. We also argue that the community should distinguish between continual learning and sequential learning with small replay buffers. Deviating from the core definition of continual learning creates a fundamentally different problem setting. These findings suggest that simply storing and replaying samples during training may not be the optimal strategy.

Following (Vahedifar & Zhang, 2026), we analyze weight pruning in the CIL scenario after learning 10 tasks to assess parameter utilization efficiency. Counter-intuitively, sharp accuracy drops under pruning indicate efficient parameter usage, most weights contribute meaningfully, while robustness to pruning reveals redundancy. The results in Fig. 3 show that both SNV and NFL+ exhibit sharp accuracy cliffs, but SNV consistently sustains its performance under deeper pruning before collapsing. On CIFAR-100, NFL+ drops from 50.1% to 12.8% between 20% and 30% pruning, whereas SNV maintains 50.5% accuracy up to 30% pruning before its cliff at 40%. This pattern holds across datasets: on Tiny-ImageNet, NFL+ collapses at $\sim 15\%$

pruning while SNV holds until $\sim 20\%$; on ImageNet-1k, the gap is $\sim 20\%$ versus $\sim 25\%$. The earlier cliff point indicates that NFL+ packs task-relevant information more densely, fewer weights are redundant, so each additional pruned weight is more likely to be critical. In contrast, PEC degrades gradually from 29.4% to 10.8% across the full pruning range on CIFAR-100, tolerating up to 80% pruning with only moderate losses. This robustness exposes PEC's fundamental inefficiency: training $|\mathcal{N}|$ separate student networks without exploiting class similarity creates massive parameter redundancy. EWC and LwF show similarly gradual degradation (12.87% $\to$ 8.5% and 19.56% $\to$ 12.2% on CIFAR-100, respectively), confirming that these methods fail to exploit available capacity fully.

### 3.2. Comparison based on the TIL Scenario

Table 1 provides results on ImageNet-1k for 10, 20, and 50 tasks for the TIL scenario. In addition, Table 2 presents results in the TIL setting across 10 tasks for CIFAR-100 and Tiny-ImageNet. TIL simplifies the prediction problem by restricting classification to classes within the identified task rather than requiring discrimination among all classes simultaneously. On CIFAR-100 (10 tasks), SNV at $c$=0.5 achieves 79.76% with BWT=0.0, outperforming every memory-based method including DyTox (73.80%, BWT=$-3.50$), DER++ (70.60%, BWT=$-6.50$), and iCaRL (70.10%, BWT=$-5.35$), all of which store 200 exemplars. On Tiny-ImageNet, SNV reaches 74.82% versus DyTox's 68.50%, a gap of over 6 points, again with zero forgetting. On ImageNet-1k, DyTox narrowly leads (59.40% vs. 57.82%), but at the cost of negative BWT ($-6.15$) and a replay buffer, while SNV maintains perfect stability. These results show that a buffer-free method can match or exceed memory-based approaches when parameter selection is done well.

*Table 2.* ACC, BWT, and PS performance comparison for TIL scenario on CIFAR-100 and Tiny-ImageNet. Best results are highlighted for buffer-free and memory-based. Memory-based methods use a fixed buffer of 200 exemplars.

| | CIFAR-100 | | | | | | Tiny-ImageNet | | | | | |
| --- | --- | --- | --- | --- | --- | --- | --- | --- | --- | --- | --- | --- |
| | 10 tasks | | | 20 tasks | | | 10 tasks | | | 20 tasks | | |
| Method | ACC↑ | BWT↑ | PS↑ | ACC↑ | BWT↑ | PS↑ | ACC↑ | BWT↑ | PS↑ | ACC↑ | BWT↑ | PS↑ |
| *Memory-free methods* | | | | | | | | | | | | |
| **SNV**, $c$=0.03 | 71.74 ±0.33 | 0.0 | 0.52 | 66.21 ±0.48 | 0.0 | 0.46 | 69.89 ±1.62 | 0.0 | 0.54 | 63.78 ±1.85 | 0.0 | 0.48 |
| **SNV**, $c$=0.05 | 74.52 ±0.31 | 0.0 | 0.54 | 68.73 ±0.44 | 0.0 | 0.48 | 73.24 ±1.07 | 0.0 | 0.56 | 66.71 ±1.32 | 0.0 | 0.50 |
| **SNV**, $c$=0.1 | 76.19 ±0.40 | 0.0 | **0.62** | 69.82 ±0.52 | 0.0 | 0.55 | 74.73 ±1.38 | 0.0 | **0.64** | 67.95 ±1.55 | 0.0 | 0.57 |
| **SNV**, $c$=0.3 | 77.89 ±0.58 | 0.0 | 0.58 | 70.65 ±0.71 | 0.0 | 0.52 | 74.38 ±1.44 | 0.0 | 0.61 | 66.94 ±1.68 | 0.0 | 0.55 |
| **SNV**, $c$=0.5 | **79.76** ±0.44 | 0.0 | 0.60 | **71.93** ±0.62 | 0.0 | 0.54 | **74.82** ±0.93 | 0.0 | 0.59 | **67.25** ±1.18 | 0.0 | 0.53 |
| WSN, $c$=0.03 | 59.65 ±0.38 | 0.0 | 0.38 | 54.38 ±0.51 | 0.0 | 0.33 | 60.72 ±1.83 | 0.0 | 0.40 | 54.90 ±2.04 | 0.0 | 0.35 |
| WSN, $c$=0.05 | 60.19 ±0.29 | 0.0 | 0.39 | 54.62 ±0.42 | 0.0 | 0.34 | 63.22 ±1.14 | 0.0 | 0.42 | 57.05 ±1.38 | 0.0 | 0.37 |
| WSN, $c$=0.1 | 61.22 ±0.49 | 0.0 | 0.41 | 55.15 ±0.62 | 0.0 | 0.36 | 61.96 ±1.61 | 0.0 | 0.41 | 55.58 ±1.82 | 0.0 | 0.36 |
| WSN, $c$=0.3 | 63.15 ±0.70 | 0.0 | 0.42 | 56.32 ±0.85 | 0.0 | 0.37 | 62.92 ±1.57 | 0.0 | 0.42 | 56.35 ±1.74 | 0.0 | 0.37 |
| WSN, $c$=0.5 | 64.00 ±0.36 | 0.0 | 0.66 | 55.82 ±0.52 | 0.0 | 0.58 | 61.06 ±1.02 | 0.0 | 0.64 | 52.41 ±1.38 | 0.0 | 0.55 |
| NFL+ | 70.68 ±2.45 | −0.35 | 0.72 | 62.14 ±3.12 | −0.52 | **0.65** | 58.21 ±4.10 | −1.04 | 0.68 | 49.85 ±3.87 | −1.35 | **0.61** |
| DCNet | 66.80 ±2.70 | −1.68 | 0.66 | 58.50 ±3.35 | −2.30 | 0.59 | 54.80 ±4.32 | −2.90 | 0.62 | 46.20 ±4.02 | −3.75 | 0.54 |
| NISPA | 62.35 ±3.10 | −3.48 | 0.62 | 54.10 ±3.65 | −5.20 | 0.54 | 53.80 ±4.42 | −5.65 | 0.59 | 45.20 ±4.18 | −8.10 | 0.51 |
| SpaceNet | 54.15 ±3.72 | −12.40 | 0.48 | 45.50 ±4.08 | −16.80 | 0.41 | 48.90 ±4.50 | −15.35 | 0.46 | 40.25 ±4.22 | −20.70 | 0.39 |
| LwF | 52.86 ±3.50 | −39.69 | 0.45 | 44.21 ±4.15 | −43.52 | 0.39 | 49.85 ±1.59 | −41.21 | 0.43 | 41.37 ±2.84 | −46.18 | 0.37 |
| SI | 51.13 ±6.25 | −45.78 | 0.47 | 42.68 ±5.73 | −49.31 | 0.41 | 48.45 ±4.87 | −52.97 | 0.49 | 40.12 ±5.21 | −57.64 | 0.43 |
| EWC | 50.59 ±1.39 | −31.64 | 0.44 | 41.85 ±2.46 | −36.21 | 0.38 | 44.19 ±3.45 | −41.85 | 0.46 | 35.74 ±4.12 | −47.32 | 0.40 |
| *Memory-based methods* | | | | | | | | | | | | |
| DyTox | 73.80 ±2.52 | −3.50 | 0.69 | 65.60 ±3.22 | −4.45 | 0.61 | 68.50 ±3.28 | −3.55 | 0.66 | 59.30 ±3.62 | −4.75 | 0.57 |
| DER++ | 70.60 ±2.72 | −6.50 | 0.40 | 62.00 ±3.50 | −7.50 | 0.34 | 67.80 ±3.72 | −3.55 | 0.43 | 59.00 ±4.15 | −4.85 | 0.37 |
| iCaRL | 70.10 ±4.08 | −5.35 | 0.43 | 61.40 ±4.42 | −6.55 | 0.37 | 66.70 ±3.02 | −6.45 | 0.41 | 58.10 ±4.00 | −7.85 | 0.35 |

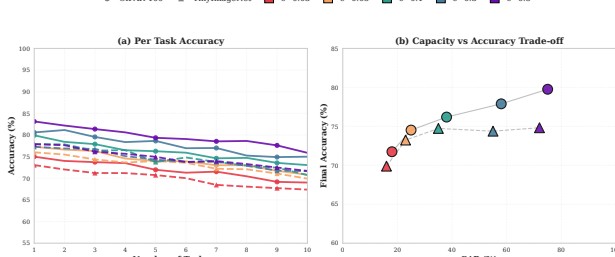

*Figure 4.* Capacity analysis on CIFAR-100 and TinyImageNet for SNV. (a) Accuracy as a function of model capacity across incremental tasks. (b) Capacity versus Accuracy.

Additionally, the comparison with WSN isolates the contribution of Shapley-based Neuron selection. Both SNV and WSN achieve BWT=0.0 across all datasets, confirming that mask-based freezing eliminates catastrophic forgetting. However, WSN reaches only 64.00% on CIFAR-100 and 61.06% on Tiny-ImageNet at the same capacity ($c$=0.5), trailing SNV by 15.76 and 13.76 points respectively. This gap widens under tighter budgets: at $c$=0.03, SNV achieves 71.74% versus WSN's 59.65% on CIFAR-100, a 12.09-point advantage.

Finding a subnetwork does not guarantee finding the *best* subnetwork; WSN's binary selection lacks a principled mechanism to rank which Neurons matter most. This motivates our approach: we leverage the strengths of both WSN and NFL+, but instead of using all parameters or freezing them blindly, we identify and select the important ones for each task. Fig. 4.(a) evaluates how model capacity affects accuracy in the TIL setting. The results indicate that capacity plays an important role in SNV, although increasing capacity does not translate into linear performance gains. For example, Fig. 4.(b), increasing capacity consistently improves performance. This naturally raises the question: *How efficiently does each method utilize the available capacity?* SNV achieves a markedly superior capacity–accuracy trade-off: using only 18–80 percent of the capacity. Additionally, across both CIFAR-100 and TinyImageNet, higher values of c generally yield improved accuracy, as each task is allocated a larger subnetwork. However, the relationship between capacity and performance is notably non-linear.

*Table 3.* Computational cost across all datasets (CIL, 10 tasks). Training time is wall-clock for the complete CL sequence. FLOPs denote total MACs. Inference latency is per image (batch = 1). Best results highlighted for buffer-free and memory-based.

| Method | CIFAR-100 | | | | | Tiny-ImageNet | | | | | ImageNet-1k | | | | |
|---|---|---|---|---|---|---|---|---|---|---|---|---|---|---|---|
| | FLOPs↓ (TF) | Train↓ Time | GPU↓ (GB) | Infer↓ (ms) | ACC↑ (%) | FLOPs↓ (TF) | Train↓ Time | GPU↓ (GB) | Infer↓ (ms) | ACC↑ (%) | FLOPs↓ (PF) | Train↓ Time | GPU↓ (GB) | Infer↓ (ms) | ACC↑ (%) |
| SI | **4,270** | **24 min** | **2.0** | **0.4** | 15.37 | **8,541** | **1.0 h** | **2.4** | **0.6** | 13.37 | **716** | **18 h** | **4.4** | **2.5** | 9.68 |
| EWC | 4,300 | 25 min | 2.3 | 0.4 | 12.87 | 8,610 | 1.1 h | 2.7 | 0.6 | 10.87 | 718 | 19 h | 4.8 | 2.5 | 7.53 |
| LwF | 5,550 | 31 min | 2.2 | 0.4 | 19.56 | 11,100 | 1.3 h | 2.6 | 0.6 | 15.56 | 930 | 23 h | 4.6 | 2.5 | 11.24 |
| SpaceNet | 5,650 | 33 min | 2.3 | 0.4 | 27.10 | 11,300 | 1.4 h | 2.7 | 0.6 | 17.80 | 947 | 25 h | 4.7 | 2.5 | 13.50 |
| PEC | 5,374 | 45 min | 2.5 | 3.2 | 29.40 | 10,748 | 1.9 h | 2.9 | 5.8 | 19.40 | 901 | 34 h | 4.9 | 18.5 | 14.83 |
| NFL | 6,735 | 37 min | 2.4 | 0.4 | 41.20 | 13,469 | 1.6 h | 2.8 | 0.6 | 32.50 | 1,129 | 28 h | 4.8 | 2.5 | 27.15 |
| NISPA | 9,100 | 1.0 h | 3.0 | 0.5 | 43.80 | 18,200 | 2.6 h | 3.4 | 0.7 | 35.40 | 1,526 | 48 h | 5.4 | 2.8 | 29.40 |
| DCNet | 6,850 | 40 min | 2.5 | 0.5 | 52.85 | 13,700 | 1.7 h | 2.9 | 0.7 | 43.90 | 1,148 | 30 h | 4.9 | 2.6 | 37.80 |
| NFL+ | 7,286 | 50 min | 2.6 | 0.5 | 53.70 | 14,573 | 2.1 h | 3.0 | 0.7 | 44.70 | 1,221 | 38 h | 5.0 | 2.7 | 38.42 |
| **SNV (ours)** | 9,035 | 62 min | 2.7 | 0.4 | **54.70** | 18,071 | 2.6 h | 3.1 | 0.6 | **45.70** | 1,514 | 47 h | 5.1 | 2.5 | **39.42** |
| *Memory-based methods* | | | | | | | | | | | | | | | |
| DER++ | **6,504** | **28 min** | 3.6 | **0.4** | 21.20 | **10,590** | **1.2 h** | 4.0 | **0.6** | 18.10 | **843** | **21 h** | 6.0 | **2.5** | 14.20 |
| iCaRL | 6,188 | 1.4 h | **3.3** | 0.8 | 21.50 | 10,275 | 3.5 h | **3.7** | 1.2 | 15.80 | 823 | 64 h | **5.7** | 4.5 | 12.45 |
| DyTox | 25,092 | 1.7 h | 5.8 | 1.9 | **55.80** | 41,356 | 4.7 h | 7.2 | 3.5 | **47.60** | 3,304 | 95 h | 14.3 | 11.2 | **40.15** |

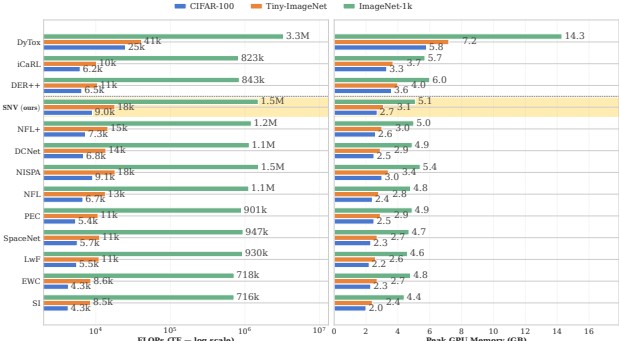

*Figure 5.* Computational cost of all compared methods across CIFAR-100, Tiny-ImageNet, and ImageNet-1k (Class-IL, 10 tasks). **Left:** total training FLOPs on a log scale. **Right:** peak GPU memory.

### 3.3. Computational Cost Comparison

A natural concern with Shapley-based importance estimation is cost: Monte Carlo sampling over Neuron coalitions adds computation that simpler proxies like Fisher information avoid entirely. Table 3 and Figure 5 quantify this overhead across all three benchmarks. The short answer is that the premium is modest. SNV requires roughly $1.24\times$ the FLOPs of NFL+ across CIFAR-100, Tiny-ImageNet, and ImageNet-1k alike, with only ∼0.1 GB additional GPU memory and no increase in inference latency. Within the buffer-free category, SNV's resource profile clusters with NFL+, NISPA, and DCNet, well below the expensive end of the spectrum occupied by DyTox. What matters, of course, is what that extra compute *buys*. The cheapest methods in the table, SI and EWC, use under half the FLOPs of SNV, yet their final accuracy collapses to single digits on ImageNet-1k. Cost comparisons are only meaningful

among methods that actually learn, and among those, SNV delivers the highest accuracy per FLOP in the buffer-free tier. Perhaps the most striking comparison is against memory-based methods. DyTox reaches $57.40\%$ on CIFAR-100, roughly three points above SNV's $54.70\%$, but at $2.8\times$ the FLOPs, $2.1\times$ the GPU memory, and with access to a stored exemplar buffer that SNV does not use. In other words, the Shapley estimation step is not a luxury; it is a lightweight investment that closes, and sometimes eliminates, the gap to replay-based methods without requiring any stored data.

## 4. Conclusion

This work introduced Shapley Neuron Values, a principled approach to continual learning that requires neither a memory buffer nor architectural expansion. SNV exploits the inherent over-parameterization of modern neural networks and leverages Shapley Values to identify and preserve the Neurons that are most critical for each task. By eliminating the need to store past data, SNV avoids the privacy risks associated with data retention and is therefore well-suited for resource-constrained and privacy-sensitive settings.

**Limitations.** SNV's main practical limitation is the cost of the post-task Shapley estimation step. This additional phase grows with network width and task complexity, which may become a bottleneck for very large models.

**Future work.** A promising direction is to eliminate the separate estimation phase for Shapley values. Recently (Wang et al., 2025a) has shown that Data Shapley values can be approximated online. We aim to adapt this to running Shapley estimates that update alongside the model parameters so that importance scores are available immediately when a task boundary is reached, without any post-hoc computation.

## Acknowledgements

This research was supported by the TOAST project, funded by the European Union's Horizon Europe research and innovation program under the Marie Skłodowska-Curie Actions Doctoral Network (Grant Agreement No. 101073465), the Danish Council for Independent Research project eTouch (Grant No. 1127- 00339B), and NordForsk Nordic University Cooperation on Edge Intelligence (Grant No. 168043).

We sincerely appreciate the reviewers for their constructive and thoughtful feedback, which made this work appealing and strong.

Dedicated to the memory of my father.

## Impact Statement

"This paper presents work whose goal is to advance the field of Machine Learning. There are many potential societal consequences of our work, none which we feel must be specifically highlighted here."

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

# A. Appendix

The Appendix mainly contains additional materials and experiments that cannot be reported due to the page limit, which is organized as follows:

- Section B contains detailed Related Work of Continual Learning approaches.

- Things We Tried That Did Not Work section C summarizes the main ideas and implementation that ultimately failed.

- The Evaluation Scenario section D outlines CL's primary scenarios used in this paper.

- The Evaluation Metrics section E provides the mathematical definitions of the metrics used in this study.

- The Additional Results section F presents FWT and PS for the TIL scenario, ACC evaluation comparison for the CIL scenario for 20 task, SNV accuracy matrix for the CIL scenario, Layer-wise Shapley Values heatmap, and Neuron mask overlap across 10 tasks for the CIL scenario.

- The Implementation details of iCaRL section G describe how the method was adapted for the TIL scenario, given that its original design specifically targets the CIL setting.

- The Hyperparameter section H provides details about the best hyperparameters selected, as well as a comprehensive list of all parameter combinations that were evaluated.

# B. Related Work

**Memory-based Continual Learning** methods maintain a buffer of past exemplars and interleave them with new task data during training. iCaRL (Rebuffi et al., 2017) selects representative samples via herding and combines them with nearest-mean classification. DER++ (Buzzega et al., 2020) stores both inputs and their corresponding logits to preserve inter-class relationships. Architecture-expanding methods offer an alternative: DyTox (Douillard et al., 2022) grows a transformer-based model with task-specific tokens, and MEMO (Zhou et al., 2023) expands deeper layers while sharing shallow features. Despite their effectiveness, memory-based methods that rely on replay buffers suffer from fundamental limitations: **i. Privacy Concerns** as storing raw samples may violate data protection regulations (e.g., GDPR (Voigt & Von dem Bussche, 2017)); **ii. Memory Overhead** since buffer size typically scales with the number of tasks, and **iii. Computational Cost** as replay mechanisms increase training time. These constraints substantially limit their applicability in scenarios with strict storage budgets or in cases where regulations prohibit exemplar storage.

**Buffer-free Continual Learning** methods operate without replay buffers and must rely on either regularization or structural constraints to prevent forgetting. EWC (Kirkpatrick et al., 2017) penalizes changes to parameters deemed important by the Fisher Information Matrix (FIM), and SI (Zenke et al., 2017) accumulates online importance estimates along the optimization trajectory. LwF (Li & Hoiem, 2018) distills knowledge from the previous model's predictions but degrades under distributional shift across tasks. More recently, SpaceNet (Sokar et al., 2021) exploits network sparsity by freeing unused capacity for new tasks, NISPA (Gurbuz & Dovrolis, 2022) combines neurogenesis-inspired growth with stability-plasticity gating in sparse networks, WSN (Kang et al., 2022) targets TIL by assigning binary masks to task-specific parameters, whereas PEC (Zajac et al., 2024) expands the network by introducing new modules per class and is limited to CIL.

DCNet (Wang et al., 2025b) maps class representations into an orthogonal hyperspherical space to achieve inter-class separation without exemplars. Additionally, the NFL (Vahedifar & Zhang, 2026) operates entirely within the fixed capacity of a standard network, relying solely on the training schedule to prevent interference without a parameter importance mechanism. While these methods eliminate the replay buffer, they typically introduce auxiliary mechanisms (sparsity masks, expanding modules, or fixed orthogonal embeddings) that constrain the model's flexibility. In contrast, the SNV operates entirely within the fixed capacity of a standard network, relying solely on an importance selection mechanism.

# C. Things We Tried That Did Not Work

In this section, we summarize several ideas and implementation strategies we explored but ultimately found ineffective. We include them here to provide transparency and to highlight directions that may appear promising but did not yield practical or stable improvements.

**1. Exact Shapley Value Computation.** We attempted to compute Shapley Values exactly rather than through estimation. Even on a small dataset such as PMNIST, this required one week of computation on four NVIDIA A6000 GPUs and completed only 30 percent of the calculation. Exact computation is therefore computationally infeasible for continual learning.

**2. Increasing Capacity Beyond 0.5.** We evaluated model capacities greater than 0.5, but did not observe any meaningful performance improvement. The additional capacity did not translate into better representations or stability.

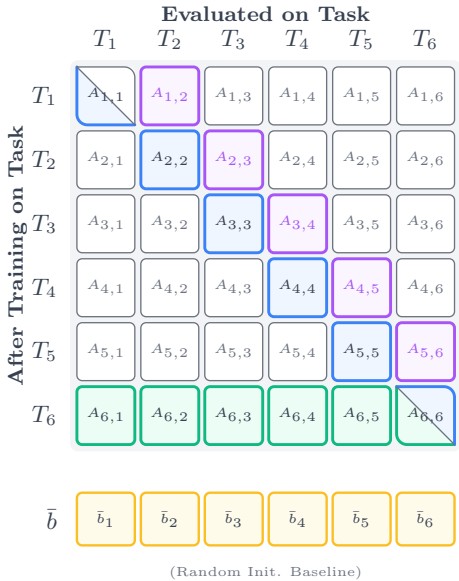

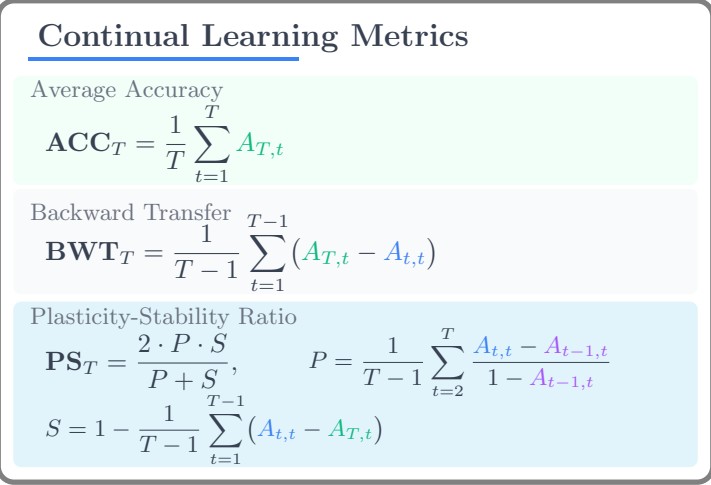

**Continual Learning Metrics**

Average Accuracy
$$\mathbf{ACC}_T = \frac{1}{T} \sum_{t=1}^{T} A_{T,t}$$

Backward Transfer
$$\mathbf{BWT}_T = \frac{1}{T-1} \sum_{t=1}^{T-1} \left( A_{T,t} - A_{t,t} \right)$$

Plasticity-Stability Ratio
$$\mathbf{PS}_T = \frac{2 \cdot P \cdot S}{P + S}, \qquad P = \frac{1}{T-1} \sum_{t=2}^{T} \frac{A_{t,t} - A_{t-1,t}}{1 - A_{t-1,t}}$$
$$S = 1 - \frac{1}{T-1} \sum_{t=1}^{T-1} \left( A_{t,t} - A_{T,t} \right)$$

*Figure 6.* Performance evaluation metrics of continual learning methods. RAC is the Random model ACcuracy.

**3. Removing the Cumulative Shapley Mask.** We experimented with training without the cumulative Shapley mask and found that performance collapsed. This confirmed that the mask is a core mechanism: without it, the network fails to properly freeze parameters associated with previous tasks.

**4. Allowing a Small Percentage of Frozen Weights to Change.** We allowed 1, 2, 3, and 5 percent of previously frozen weights to update. Even these small relaxations caused large drops in accuracy, demonstrating the sensitivity of the model to parameter leakage across tasks.

**5. Disallowing New Tasks from Accessing Frozen Parameters.** In early experiments, we fully restricted new tasks from using any frozen parameters. This led to substantially worse performance than our final approach, showing that some shared reuse of stabilized parameters is necessary.

**6. Restricting the Memory Buffer Size.** We also tested aggressive reductions in the memory buffer size. Performance degraded sharply, reinforcing that, even for buffer-free baselines, some degree of replay is essential in hybrid settings.

## D. Evaluation Scenario

The two main experimental scenarios typically used to evaluate the performance of methods are the following:

**1. Task Incremental Learning (TIL):** In TIL, the training data is divided into multiple tasks, each with a unique set of classes. The crucial aspect of TIL is that the model is provided with information about which task it is handling during training and testing. This allows the model to use the computational graph corresponding to each task. For example, if the model is trained to classify images of animals and vehicles, the task label information is also provided for testing on a new image; thus, the network's classification output for the corresponding task will be calculated. This knowledge simplifies the inference task, as the model does not need to consider all possible classes simultaneously (Wickramasinghe et al., 2024; Vahedifar et al., 2026).

**2. Class Incremental Learning (CIL):** In CIL, the model is also trained on different tasks, but is not told which task a new sample belongs to during testing. Instead, regardless of the task, the model needs to respond to all the classes it has encountered. This makes CIL more challenging than TIL, as the model must infer the correct class without task-related information. For instance, after training a model to recognize animals and vehicles separately, CIL would test the model on all classes simultaneously (animals and vehicles) without informing the model whether it is currently classifying an animal or a vehicle (Qu et al., 2025; Zhou et al., 2024a).

Physically, our model utilizes a single output layer matrix $W_{\mathrm{out}} \in \mathbb{R}^{(C_{\mathrm{old}} + C_{\mathrm{new}}) \times F}$, where F is the feature dimension. Task identifiers are strictly a *training-time* requirement used to define the boundaries of these logical partitions for loss computation. During inference, particularly in the CIL set-

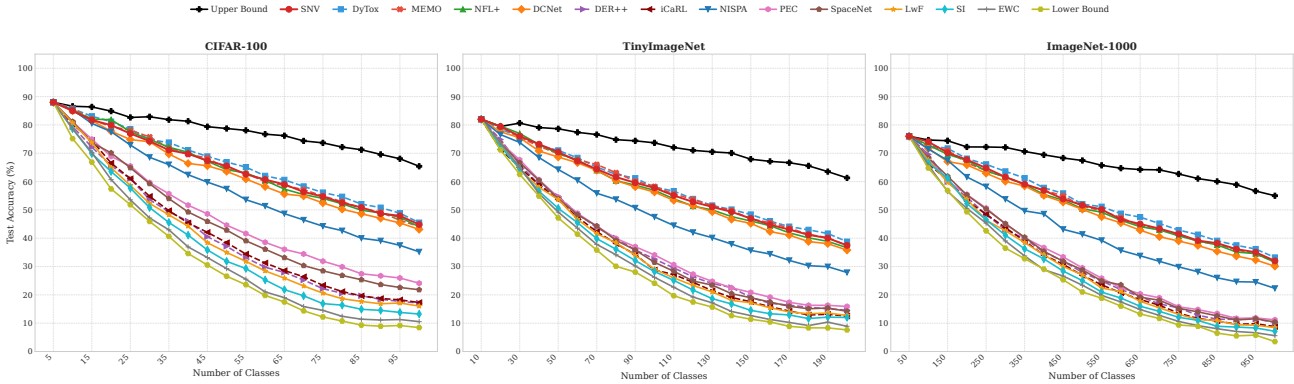

*Figure 7.* ACC evaluation comparison for the CIL for 20 tasks for each dataset. Each point represents the average classification accuracy evaluated after learning a given task, averaged over all tasks learned up to that point. For example, the value at task 10 corresponds to the average accuracy of the model on the test sets of tasks 1 through 10 after completing training on task 10. For details, see Fig. 9. Solid lines: buffer-free; dashed lines: memory-based.

ting, the partitions are ignored, and the single head functions as a global classifier over the union of all classes.

## E. Evaluation Metrices

To evaluate continual learning methods, we measure the model's accuracy on all tasks after training on each sequential task $T_t$ using dataset $\mathcal{D}_t$. This produces an accuracy matrix $A \in \mathbb{R}^{T \times T}$, where element $A_{i,j}$ represents the model's accuracy on task $T_j$ after completing training on task $T_i$. Figure 6 illustrates the most common metrics in continual learning and how these metrics are computed from the accuracy matrix.

In addition, we define **Capacity (CAP)**, which measures the total memory footprint of the continual learning model relative to the dense backbone network. Since our method utilizes structured pruning at the Neuron level, the storage overhead for task-specific binary masks is negligible (orders of magnitude smaller than the model weights). Therefore, unlike previous weight-level pruning methods (Kang et al., 2022) that require complex mask compression terms, we define Capacity simply as the percentage of unique network parameters activated by the union of all learned tasks:

$$\text{CAP} = \frac{\| \bigcup_{t=1}^{T} \mathcal{M}^* \|_0}{|\theta_{dense}|} \times 100\% \quad (12)$$

where $\| \cdot \|_0$ represents the count of non-zero parameters in the union of all task-specific subnetworks, and $|\theta_{dense}|$ is the total parameter count of the dense backbone. A CAP of 100% implies the model utilizes the full memory capacity of the standard network, while a lower CAP indicates efficient parameter reuse and sparsity.

In summary, an effective continual learning method should maximize ACC, BWT, and PS. When comparing methods with similar ACC values, those with higher PS and BWT

are preferable. Note that BWT for the first task is undefined (Lopez-Paz & Ranzato, 2017).

## F. Additional Results

This section provides additional experimental results excluded from the main paper due to page limitations.

### F.1. Task Granularity Analysis

Comparing the results in Fig. 2 with those in Fig. 7 reveals that increasing the number of tasks leads to clear performance degradation, and this decline is substantially larger than the drop observed in the upper bound. This indicates that as the number of tasks grows, continual learning methods struggle to adequately prevent catastrophic forgetting.

Another notable finding arises from limiting the memory buffer for memory-based methods to 200 exemplars in this analysis. Under this constraint, the performance gap between NFL+, SNV, and other memory-based approaches such as DER++ and iCaRL becomes much more noticeable. This suggests that memory-based methods rely heavily on sufficiently large replay buffers, and that memory alone is not always the solution. DyTox, due to its dynamic architecture, is more resilient to restricted memory, yet its performance remains comparable to SNV in both the 10-task and 20-task settings. These observations raise several important questions for the community.

*How much memory should we allocate to memory-based methods? And what should be the composition of the memory buffer?*

*Are these truly continual learning methods, or do they simply perform sequential learning with replay?*

*Is it fair to compare memory-based and buffer-free ap-*

*Table 4.* Performance comparison on CIFAR-100 and Tiny-ImageNet across 10 and 20 tasks for CIL scenario. Best results are highlighted for buffer-free and memory-based.

| | CIFAR-100 | | | | | | TinyImageNet | | | | | |
| | 10 tasks | | | 20 tasks | | | 10 tasks | | | 20 tasks | | |
| Method | ACC↑ | BWT↑ | PS↑ | ACC↑ | BWT↑ | PS↑ | ACC↑ | BWT↑ | PS↑ | ACC↑ | BWT↑ | PS↑ |
|---|---|---|---|---|---|---|---|---|---|---|---|---|
| *Memory-free methods* | | | | | | | | | | | | |
| **SNV (ours)** | **54.70** ± 0.42 | **−0.04** | **0.69** | **44.85** ± 0.55 | **−0.04** | **0.61** | **45.70** ± 1.15 | **−0.05** | **0.62** | **37.47** ± 1.38 | **−0.04** | **0.54** |
| NFL+ | 53.70 ± 2.45 | −0.05 | 0.67 | 44.03 ± 3.12 | −0.05 | 0.60 | 44.70 ± 3.82 | −0.06 | 0.63 | 36.65 ± 4.05 | −0.07 | 0.56 |
| DCNet | 52.85 ± 2.70 | −0.06 | 0.65 | 43.10 ± 3.35 | −0.06 | 0.58 | 43.90 ± 4.10 | −0.07 | 0.60 | 35.80 ± 4.22 | −0.08 | 0.53 |
| NISPA | 43.80 ± 3.10 | −0.10 | 0.58 | 35.20 ± 3.65 | −0.10 | 0.52 | 35.40 ± 4.42 | −0.12 | 0.54 | 27.90 ± 4.58 | −0.12 | 0.47 |
| SpaceNet | 27.10 ± 3.72 | −0.15 | 0.48 | 21.80 ± 4.08 | −0.13 | 0.44 | 17.80 ± 4.50 | −0.18 | 0.43 | 14.20 ± 4.35 | −0.16 | 0.40 |
| PEC | 29.40 ± 4.25 | −0.19 | 0.52 | 24.11 ± 3.95 | −0.15 | 0.48 | 19.40 ± 4.68 | −0.22 | 0.48 | 15.91 ± 4.42 | −0.17 | 0.42 |
| LwF | 19.56 ± 3.50 | −0.18 | 0.40 | 16.04 ± 4.15 | −0.16 | 0.38 | 15.56 ± 1.59 | −0.20 | 0.39 | 12.76 ± 2.84 | −0.17 | 0.39 |
| SI | 15.37 ± 6.25 | −0.25 | 0.42 | 13.23 ± 5.73 | −0.20 | 0.41 | 13.37 ± 4.87 | −0.27 | 0.47 | 12.02 ± 5.21 | −0.22 | 0.43 |
| EWC | 12.87 ± 1.39 | −0.25 | 0.39 | 10.55 ± 2.46 | −0.20 | 0.37 | 10.87 ± 3.45 | −0.26 | 0.44 | 8.91 ± 4.12 | −0.22 | 0.40 |
| *Memory-based methods* | | | | | | | | | | | | |
| DyTox | **57.40** ± 2.52 | **−0.37** | **0.74** | **47.07** ± 3.22 | **−0.30** | **0.65** | **49.40** ± 3.28 | **−0.39** | **0.70** | **40.51** ± 3.62 | **−0.32** | **0.61** |
| iCaRL | 25.41 ± 4.08 | −0.71 | 0.44 | 20.84 ± 4.42 | −0.58 | 0.40 | 18.41 ± 3.02 | −0.73 | 0.41 | 15.10 ± 4.00 | −0.60 | 0.37 |
| DER++ | 25.30 ± 2.72 | −0.72 | 0.42 | 20.75 ± 3.50 | −0.59 | 0.39 | 21.30 ± 3.72 | −0.71 | 0.44 | 17.47 ± 4.15 | −0.58 | 0.40 |

*Figure 8.* ACC matrix for SNV for the CIL for 10 tasks for each dataset.

*proaches under current evaluation protocols?*

*Are memory-based approaches even applicable in real-world scenarios where privacy constraints, data retention policies, or legal regulations restrict storing past examples?*

### F.2. CIL Analysis for CIFAR-100 and Tiny-ImageNet

Table 4 summarizes performance on CIFAR-100 and Tiny-ImageNet under both 10- and 20-task Class-IL splits. Three observations stand out. First, SNV leads every buffer-free method on every metric in every setting. On CIFAR-100 with 10 tasks, it reaches 54.70%, one point above NFL+ and nearly two points above DCNet, while maintaining a BWT of −0.04. The gap widens as the number of tasks grows: at 20 tasks, SNV still holds 44.85%, whereas NFL+ drops to 44.03% and DCNet to 43.10%, both with noticeably worse backward transfer. The same pattern repeats

on Tiny-ImageNet, confirming that the advantage is not dataset-specific.

Second, the variance tells an important story. SNV's standard deviations are consistently the smallest in the buffer-free group (±0.42 on CIFAR-100/10 tasks versus ±2.45 for NFL+ and ±2.70 for DCNet). Shapley-based importance estimation evidently produces more stable Neuron selections across random seeds than the heuristics used by competing methods, which is a practical benefit that raw accuracy alone does not capture.

Third, SNV closes most of the gap to memory-based methods without storing a single exemplar. DyTox, the strongest replay method, reaches 57.40% on CIFAR-100 at 10 tasks, only 2.7 points above SNV, yet it does so with a BWT of −0.37, an order of magnitude worse than SNV's −0.04. In other words, DyTox buys its accuracy at the cost of sub-

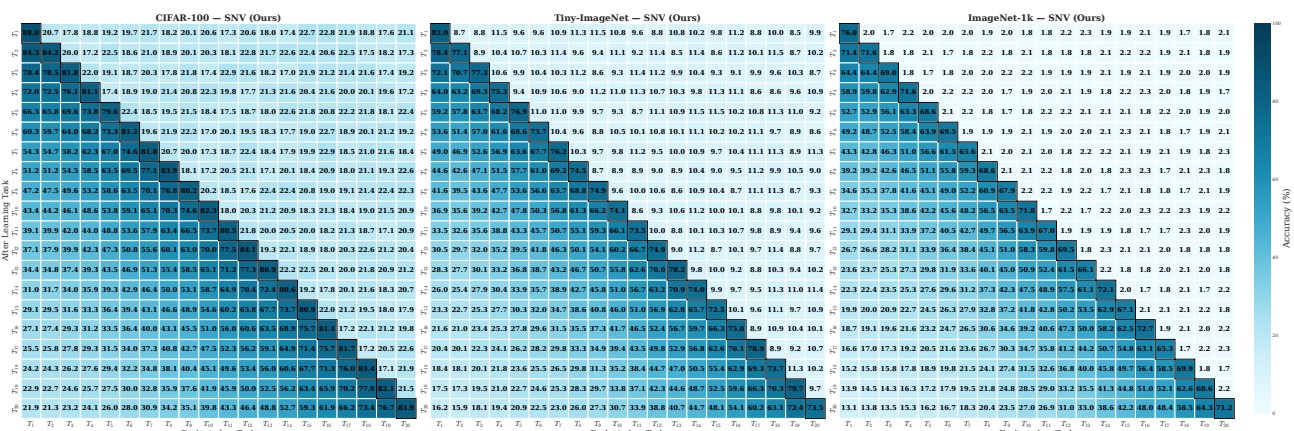

*Figure 9.* ACC matrix for SNV for the CIL for 20 tasks for each dataset.

stantial forgetting that a replay buffer must continuously patch. On Tiny-ImageNet the story is similar: DyTox leads by 3.7 points in accuracy but trails by 0.34 in backward transfer. For any deployment where data retention is restricted, whether by privacy regulation, memory constraints, or both, SNV offers a compelling trade-off: nearly equivalent accuracy with essentially no forgetting and no stored data.

### F.3. ACC matrix for SNV

The ACC matrices in Fig. 8 and Fig. 9 reveal a clear pattern: catastrophic forgetting intensifies as both the number of tasks and dataset complexity increase. In the 10-task CIFAR-100 matrix, Task 1 accuracy degrades from 90.0% (after learning $T_1$) to 33.5% (after learning T10), representing a 56.5% point drop. However, in the 20-task CIFAR-100 scenario, Task 1 accuracy plummets from 88.0% to 21.9%, a 66.1% drop spread over twice the training horizon. TinyImageNet exhibits even more severe forgetting: the 10-task final $T_1$ accuracy is 26.7%, while the 20-task version reaches just 16.2%.

A striking observation across all matrices is the non-linear forgetting trajectory; early tasks experience rapid accuracy degradation in the first few subsequent tasks, then stabilize somewhat as training progresses. For instance, in the 20-task ImageNet-1k matrix, $T_1$ drops from 76.0% → 71.4% → 64.4% in the first three steps (steep decline), but the later decrements become more gradual. The diagonal entries (representing plasticity) remain consistently high across all scenarios, regardless of task count, indicating SNV preserves strong learning capacity. The upper-triangular values hover near a random baseline, confirming minimal positive transfer in class-incremental settings where future classes haven't been seen.

### F.4. Shapley Value Heatmap Analysis

The Shapley value heatmaps in Fig. 10 provide a diagnostic visualization of Neuron importance distribution across tasks and network layers. This analysis reveals the core mechanism underlying SNV's selective parameter protection strategy. The heatmaps exhibit pronounced sparsity, with the majority of Neurons displaying low Shapley Values (white/light green regions) while a minority demonstrate high importance (dark green). This sparsity increases progressively with task index, reflecting the cumulative "Freezing" of critical Neurons as training advances. Mathematically, if we denote the sparsity ratio at task $t$ as $\rho_t$, we observe $\rho_t > \rho_{t-1}$, indicating that fewer Neurons remain available for plasticity as the task sequence progresses.

Notably, Certain Neurons maintain consistently high Shapley Values across all tasks, appearing as vertical streaks in the heatmap. These Neurons encode generalizable features, such as edge detectors and texture patterns, that benefit multiple tasks simultaneously. For ResNet-18 architectures, the visualization spans eight BasicBlocks with channel dimensions $\{64, 64, 128, 128, 256, 256, 512, 512\}$. Early layers (layer1.x) typically exhibit more uniform importance distributions, as they capture low-level features shared across tasks. In contrast, later layers (layer4.x) demonstrate greater task-specific variability, reflecting their role in encoding discriminative, class-specific representations.

### F.5. Task Mask Overlap Analysis

The mask overlap heatmap in Fig. 11, quantifies the degree of Neuron sharing across tasks using the Jaccard coefficient, defined as $J(M_i, M_j) = |M_i \cap M_j| / |M_i \cup M_j|$, where $M_i$ and $M_j$ denote the binary importance masks for tasks $T_i$ and $T_j$, respectively. This metric ranges from 0 (completely disjoint Neuron subsets) to 1 (identical masks), with diagonal entries trivially equal to unity. The off-diagonal values

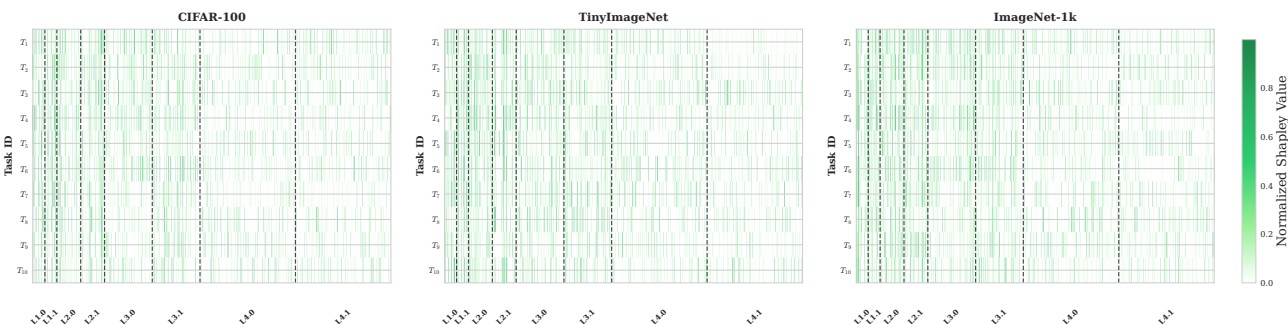

*Figure 10.* Layer-wise Shapley Neuron Importance Across Datasets.

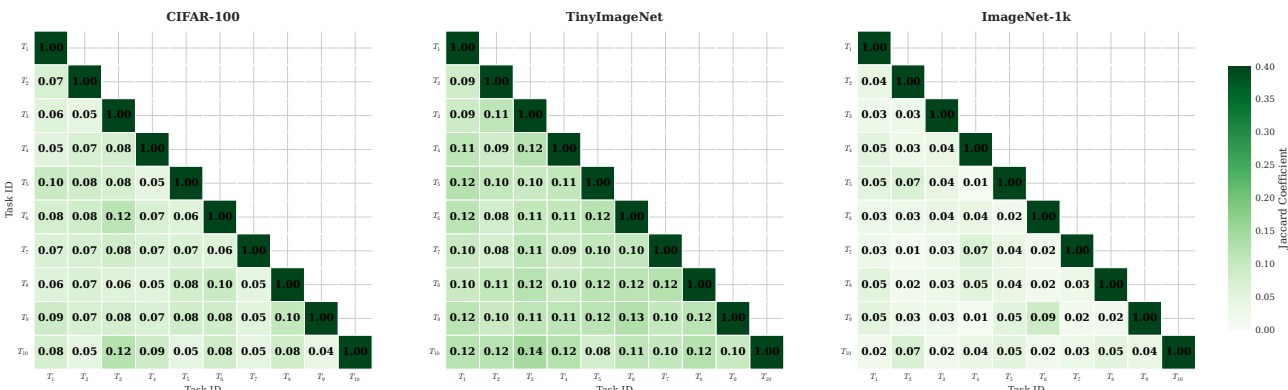

*Figure 11.* Mask overlap between tasks across datasets for 10 tasks in the CIL scenario.

reveal critical insights into SNV's Neuron allocation strategy: moderate overlap values (typically 0.3–0.5) indicate that SNV achieves an effective balance between *knowledge sharing* and *interference avoidance*. Shared Neurons facilitate positive forward transfer by reusing generalizable features across tasks, while distinct Neuron subsets prevent catastrophic forgetting by isolating task-specific representations. Excessively high overlap would indicate that new tasks overwrite previously important Neurons, leading to severe forgetting; conversely, excessively low overlap would suggest inefficient capacity utilization and limited transfer.

### F.6. Wall-clock breakdown

A common concern with Shapley-based methods is computational cost. Tables 5 and 6 address this directly. The per-transition valuation overhead is modest: on CIFAR-100 it adds 1.3 minutes on top of a 5-minute training phase, and on Tiny-ImageNet 4.2 minutes on top of 11.8 minutes (Table 5). The bulk of this time is spent on Monte Carlo forward passes; truncation and multi-armed bandit (MAB) sampling together account for less than a tenth of one task's training cost. Over a full 10-task sequence, the valuation phase totals 11.7 minutes for CIFAR-100 and 37.8 minutes for Tiny-ImageNet, roughly 23% and 24% of the overall runtime, respectively. For comparison, EWC completes

faster but reaches only 12.87% accuracy on CIFAR-100 versus SNV's 54.70%.

A natural question is how much accuracy the approximation sacrifices. Table 6 answers this by peeling back the acceleration layers one at a time. Pure Monte Carlo sampling (row a) reaches 54.92% but takes 97 minutes. Adding truncation (row b) halves the per-transition cost with only a 0.08 point accuracy drop. The full TMAB-Shapley pipeline (row c) halves it again, 4× faster than the baseline, while losing just 0.22 points overall. The Shapley score standard deviation does rise from 0.008 to 0.018, but this has a negligible impact on Neuron ranking: the importance ordering remains stable across all five seeds, as reflected in the tight accuracy variance (±0.23). In short, the speedup is nearly free.

## G. Implementation Details of iCaRL

We adapt the iCaRL method (Rebuffi et al., 2017), originally designed for CIL, to the TIL setting by modifying its classification strategy.

In its standard formulation, iCaRL assigns a label $y^*$ by computing the nearest neighbor between the input's feature representation and the mean exemplar features of each class. Let $\overline{\mathbf{y}}$ denote the mean feature vector computed from stored exemplars of class $y$, and let $\phi(\mathbf{x})$ represent the feature

*Table 5.* Per-task wall-clock breakdown for SNV. Task 1 requires training only; tasks 2–10 additionally run the TMAB-Shapley valuation phase.

| Phase | CIFAR-100 | | TinyImageNet | |
|---|---|---|---|---|
| | Frac. | Time | Frac. | Time |
| Task training (per task) | 1.00× | 5.0 min | 1.00× | 11.8 min |
| *Valuation overhead (per transition, tasks 2–10 only):* | | | | |
| MC forward passes | 0.18× | 54 s | 0.25× | 3.0 min |
| Truncation + MAB | 0.08× | 24 s | 0.10× | 1.2 min |
| Gradient mask update | 0.01× | 3 s | 0.01× | 8 s |
| **Valuation total** | **0.26×** | **1.3 min** | **0.36×** | **4.2 min** |
| *Full CL sequence:* | | | | |
| Training (10 tasks) | | 50.0 min | | 118 min |
| Valuation (9 transitions) | | 11.7 min | | 37.8 min |
| **SNV total** | | **62 min** | | **2.6 h** |
| EWC total (for reference) | | 27 min | | 1.1 h |

Frac.: relative to one task's training time.

*Table 6.* Ablation study on TMAB-Shapley approximation layers (CIFAR-100, CIL, 10 tasks).

| Approximation Variant | Valuation / trans. | Total Time | ACC (%) | ACC std. | Shapley std. |
|---|---|---|---|---|---|
| (a) MC-only | 5.2 min | 97 min | 54.92 | ±0.18 | 0.008 |
| (b) MC + Truncation | 2.6 min | 73 min | 54.84 | ±0.21 | 0.012 |
| (c) **Full TMAB-Shapley** | **1.3 min** | **62 min** | **54.70** | **±0.23** | **0.018** |

*Valuation / trans.*: wall-clock per task transition (tasks 2–10 only). *Total Time*: full 10-task CL sequence (training + valuation). *Shapley std.*: standard deviation of normalized Neuron importance scores across runs (lower = more stable). The ∼0.22% accuracy gap between (a) and (c) confirms that MAB acceleration and truncation preserve estimation quality while delivering a 4× speedup.

embedding of input **x**. The original iCaRL classification rule is:

$$y^* = \underset{y=1,...,t}{\operatorname{argmin}} \|\phi(\mathbf{x}) - \overline{\mathbf{y}}\|_2. \qquad (13)$$

We reformulate this as a scoring function $h(\mathbf{x})$ that computes negative Euclidean distances between the input features and a matrix **Φ** containing all class prototype vectors:

$$h(\mathbf{x}) = -\|\phi(\mathbf{x}) - \mathbf{\Phi}\|_2. \qquad (14)$$

This reformulation preserves equivalence with Eq. 13 under CIL evaluation, as maximizing $h(\mathbf{x})$ produces identical predictions to minimizing the distance in the original formulation.

## H. Hyperparameter Search

We present the complete hyperparameter search space in Table 7 and the selected best hyperparameter combinations for each method in Table 8. We denote the learning rate

with $lr$, weight decay with $wd$, and Adam optimizer hyperparameters with $\beta_1$, $\beta_2$, and $\epsilon$. For NFL+ Auto-Encoder training, we use separate Adam parameters denoted with $lr_{adam}$. All methods are optimized using the Adam optimizer with default parameters: $\beta_1 = 0.9$, $\beta_2 = 0.999$, $\epsilon = 10^{-8}$.

This first-task HPO strategy avoids the two principal pitfalls identified in prior work: (i) the conventional protocol of tuning and evaluating within the same scenario, which Cha & Cho (2025) show leads to significant overestimation of CL capacity across more than 8,000 experiments; and (ii) end-of-training HPO, which Lee et al. (2024) demonstrates is unrealistic because it requires repeated passes over the entire task stream. We emphasize that this concern disproportionately affects memory-based methods, whose performance depends critically on buffer management hyperparameters (e.g., reservoir sampling rate, replay frequency, exemplar selection strategy). Tuning these parameters on the evaluation scenario implicitly leaks information about the distribution of future tasks into the buffer policy, precisely the information leakage that Cha & Cho (2025) identify as the root cause of overestimation. NFL+ is *structurally exempt* from this failure mode: as a buffer-free method, it maintains no exemplar buffer and therefore has no buffer management hyperparameters to tune. Its hyperparameters govern the optimization dynamics rather than data selection and are determined entirely by the first task.

*Table 7.* Complete hyperparameter search space for all methods across datasets. All methods use the Adam optimizer with shared parameters $\beta_1 \in \{0.9\}$, $\beta_2 \in \{0.999\}$, $\epsilon \in \{10^{-8}\}$. For CIL, memory-based methods use a buffer of 2,000 (CIFAR-100, Tiny-ImageNet) or 20,000 (ImageNet-1k). For TIL, the buffer is fixed at 200.

| Method | Scenario | Dataset | Hyperparameters (searched on first task only) | | | |
|---|---|---|---|---|---|---|
| **SNV (ours)** | CIL/TIL | CIFAR-100
Tiny-ImageNet
ImageNet-1k | $lr$: [0.0001, 0.001, 0.01, 0.03, 0.1],
$lr$: [0.0001, 0.001, 0.01, 0.03, 0.1],
$lr$: [0.0001, 0.001, 0.01, 0.03, 0.1], | $c$: [0.05, 0.1, 0.2],
$c$: [0.05, 0.1, 0.2],
$c$: [0.05, 0.1, 0.2], | $\tau$: [0.01, 0.05, 0.1],
$\tau$: [0.01, 0.05, 0.1],
$\tau$: [0.01, 0.05, 0.1], | $\alpha$: [0.90, 0.95, 0.99]
$\alpha$: [0.90, 0.95, 0.99]
$\alpha$: [0.90, 0.95, 0.99] |
| NFL+ | CIL/TIL | CIFAR-100

Tiny-ImageNet

ImageNet-1k | $lr$: [0.0001, 0.001, 0.01, 0.03, 0.1],
$lr_{\mathrm{ae}}$: [0.0001, 0.001, 0.01]
$lr$: [0.0001, 0.001, 0.01, 0.03, 0.1],
$lr_{\mathrm{ae}}$: [0.0001, 0.001, 0.01]
$lr$: [0.0001, 0.001, 0.01, 0.03, 0.1],
$lr_{\mathrm{ae}}$: [0.0001, 0.001, 0.01] | $p$: [2.0, 3.0, 4.0],

$p$: [2.0, 3.0, 4.0],

$p$: [2.0, 3.0, 4.0], | $\Omega$: [0.1, 0.3, 0.5, 1.0],

$\Omega$: [0.1, 0.3, 0.5, 1.0],

$\Omega$: [0.1, 0.3, 0.5, 1.0], | $\eta$: [0.3, 0.5, 0.7]

$\eta$: [0.3, 0.5, 0.7]

$\eta$: [0.3, 0.5, 0.7] |
| DCNet | CIL/TIL | CIFAR-100
Tiny-ImageNet
ImageNet-1k | $lr$: [0.0001, 0.001, 0.01, 0.03, 0.1],
$lr$: [0.0001, 0.001, 0.01, 0.03, 0.1],
$lr$: [0.0001, 0.001, 0.01, 0.03, 0.1], | $\lambda_{\mathrm{dc}}$: [0.1, 0.5, 1.0, 5.0]
$\lambda_{\mathrm{dc}}$: [0.1, 0.5, 1.0, 5.0]
$\lambda_{\mathrm{dc}}$: [0.1, 0.5, 1.0, 5.0] | | |
| NISPA | CIL/TIL | CIFAR-100
Tiny-ImageNet
ImageNet-1k | $lr$: [0.0001, 0.001, 0.01, 0.03, 0.1],
$lr$: [0.0001, 0.001, 0.01, 0.03, 0.1],
$lr$: [0.0001, 0.001, 0.01, 0.03, 0.1], | $s$: [0.3, 0.5, 0.7, 0.9],
$s$: [0.3, 0.5, 0.7, 0.9],
$s$: [0.3, 0.5, 0.7, 0.9], | $\lambda_{\mathrm{reg}}$: [0.1, 1.0, 10.0]
$\lambda_{\mathrm{reg}}$: [0.1, 1.0, 10.0]
$\lambda_{\mathrm{reg}}$: [0.1, 1.0, 10.0] | |
| WSN | TIL | CIFAR-100
Tiny-ImageNet
ImageNet-1k | $lr$: [0.0001, 0.001, 0.01, 0.03]
$lr$: [0.0001, 0.001, 0.01, 0.03]
$lr$: [0.0001, 0.001, 0.01, 0.03] | | | |
| PEC | CIL/TIL | CIFAR-100
Tiny-ImageNet
ImageNet-1k | $lr$: [0.0001, 0.001, 0.01, 0.03],
$lr$: [0.0001, 0.001, 0.01, 0.03],
$lr$: [0.0001, 0.001, 0.01, 0.03], | $\lambda_{\mathrm{pec}}$: [0.1, 0.5, 1.0, 5.0]
$\lambda_{\mathrm{pec}}$: [0.1, 0.5, 1.0, 5.0]
$\lambda_{\mathrm{pec}}$: [0.1, 0.5, 1.0, 5.0] | | |
| SpaceNet | CIL/TIL | CIFAR-100
Tiny-ImageNet
ImageNet-1k | $lr$: [0.0001, 0.001, 0.01, 0.03],
$lr$: [0.0001, 0.001, 0.01, 0.03],
$lr$: [0.0001, 0.001, 0.01, 0.03], | $s_{\mathrm{init}}$: [0.3, 0.5, 0.7],
$s_{\mathrm{init}}$: [0.3, 0.5, 0.7],
$s_{\mathrm{init}}$: [0.3, 0.5, 0.7], | $\lambda_{\mathrm{sp}}$: [0.1, 1.0, 10.0]
$\lambda_{\mathrm{sp}}$: [0.1, 1.0, 10.0]
$\lambda_{\mathrm{sp}}$: [0.1, 1.0, 10.0] | |
| LwF | CIL/TIL | CIFAR-100
Tiny-ImageNet
ImageNet-1k | $lr$: [0.0001, 0.001, 0.01, 0.03, 0.1],
$lr$: [0.0001, 0.001, 0.01, 0.03, 0.1],
$lr$: [0.0001, 0.001, 0.01, 0.03, 0.1], | $\alpha$: [0.3, 0.5, 1.0, 3.0],
$\alpha$: [0.3, 0.5, 1.0, 3.0],
$\alpha$: [0.3, 0.5, 1.0, 3.0], | $T$: [2.0, 4.0],
$T$: [2.0, 4.0],
$T$: [2.0, 4.0], | $wd$: [1e-5, 5e-5]
$wd$: [1e-5, 5e-5]
$wd$: [1e-5, 5e-5] |
| EWC | CIL/TIL | CIFAR-100
Tiny-ImageNet
ImageNet-1k | $lr$: [0.0001, 0.001, 0.01, 0.03, 0.1],
$lr$: [0.0001, 0.001, 0.01, 0.03, 0.1],
$lr$: [0.0001, 0.001, 0.01, 0.03, 0.1], | $\lambda$: [10, 25, 30, 90, 100],
$\lambda$: [10, 25, 30, 90, 100],
$\lambda$: [10, 25, 30, 90, 100], | $\gamma$: [0.9, 0.95, 1.0]
$\gamma$: [0.9, 0.95, 1.0]
$\gamma$: [0.9, 0.95, 1.0] | |
| SI | CIL/TIL | CIFAR-100
Tiny-ImageNet
ImageNet-1k | $lr$: [0.0001, 0.001, 0.01, 0.03, 0.1],
$lr$: [0.0001, 0.001, 0.01, 0.03, 0.1],
$lr$: [0.0001, 0.001, 0.01, 0.03, 0.1], | $c_{SI}$: [0.3, 0.5, 0.7, 1.0],
$c_{SI}$: [0.3, 0.5, 0.7, 1.0],
$c_{SI}$: [0.3, 0.5, 0.7, 1.0], | $\xi$: [0.9, 1.0]
$\xi$: [0.9, 1.0]
$\xi$: [0.9, 1.0] | |
| DyTox | CIL/TIL | CIFAR-100
Tiny-ImageNet
ImageNet-1k | $lr$: [0.0001, 0.001, 0.01, 0.03, 0.1]
$lr$: [0.0001, 0.001, 0.01, 0.03, 0.1]
$lr$: [0.0001, 0.001, 0.01, 0.03, 0.1] | | | |
| iCaRL | CIL/TIL | CIFAR-100
Tiny-ImageNet
ImageNet-1k | $lr$: [0.0001, 0.001, 0.01, 0.03, 0.1],
$lr$: [0.0001, 0.001, 0.01, 0.03, 0.1],
$lr$: [0.0001, 0.001, 0.01, 0.03, 0.1], | $wd$: [0, 1e-5, 5e-5, 1e-4]
$wd$: [0, 1e-5, 5e-5, 1e-4]
$wd$: [0, 1e-5, 5e-5, 1e-4] | | |
| DER++ | CIL/TIL | CIFAR-100
Tiny-ImageNet
ImageNet-1k | $lr$: [0.0001, 0.001, 0.01, 0.03, 0.1],
$lr$: [0.0001, 0.001, 0.01, 0.03, 0.1],
$lr$: [0.0001, 0.001, 0.01, 0.03, 0.1], | $\alpha$: [0.1, 0.2, 0.3, 0.5, 1.0],
$\alpha$: [0.1, 0.2, 0.3, 0.5, 1.0],
$\alpha$: [0.1, 0.2, 0.3, 0.5, 1.0], | $\beta$: [0.5, 1.0]
$\beta$: [0.5, 1.0]
$\beta$: [0.5, 1.0] | |

*Table 8.* Selected best hyperparameters for all methods across datasets.

| Method | Scenario | Dataset | Selected Hyperparameters |
|---|---|---|---|
| **SNV (ours)** | CIL/TIL | CIFAR-100 | $lr$: 0.01, $c$: 0.1, $\tau$: 0.05, $\alpha$: 0.95 |
| | | Tiny-ImageNet | $lr$: 0.01, $c$: 0.1, $\tau$: 0.05, $\alpha$: 0.95 |
| | | ImageNet-1k | $lr$: 0.001, $c$: 0.2, $\tau$: 0.05, $\alpha$: 0.95 |
| NFL+ | CIL/TIL | CIFAR-100 | $lr$: 0.01, $p$: 2.0, $\Omega$: 0.5, $\eta$: 0.5, $lr_{\mathrm{ae}}$: 0.001 |
| | | Tiny-ImageNet | $lr$: 0.01, $p$: 2.0, $\Omega$: 0.5, $\eta$: 0.5, $lr_{\mathrm{ae}}$: 0.001 |
| | | ImageNet-1k | $lr$: 0.001, $p$: 2.0, $\Omega$: 0.5, $\eta$: 0.5, $lr_{\mathrm{ae}}$: 0.001 |
| DCNet | CIL/TIL | CIFAR-100 | $lr$: 0.01, $\lambda_{\mathrm{dc}}$: 1.0 |
| | | Tiny-ImageNet | $lr$: 0.01, $\lambda_{\mathrm{dc}}$: 1.0 |
| | | ImageNet-1k | $lr$: 0.001, $\lambda_{\mathrm{dc}}$: 0.5 |
| NISPA | CIL/TIL | CIFAR-100 | $lr$: 0.01, $s$: 0.5, $\lambda_{\mathrm{reg}}$: 1.0 |
| | | Tiny-ImageNet | $lr$: 0.01, $s$: 0.5, $\lambda_{\mathrm{reg}}$: 1.0 |
| | | ImageNet-1k | $lr$: 0.001, $s$: 0.5, $\lambda_{\mathrm{reg}}$: 1.0 |
| WSN | TIL | CIFAR-100 | $lr$: 0.001 |
| | | Tiny-ImageNet | $lr$: 0.001 |
| | | ImageNet-1k | $lr$: 0.0001 |
| PEC | CIL | CIFAR-100 | $lr$: 0.001, $\lambda_{\mathrm{pec}}$: 1.0 |
| | | Tiny-ImageNet | $lr$: 0.001, $\lambda_{\mathrm{pec}}$: 1.0 |
| | | ImageNet-1k | $lr$: 0.0001, $\lambda_{\mathrm{pec}}$: 0.5 |
| SpaceNet | CIL/TIL | CIFAR-100 | $lr$: 0.001, $s_{\mathrm{init}}$: 0.5, $\lambda_{\mathrm{sp}}$: 1.0 |
| | | Tiny-ImageNet | $lr$: 0.001, $s_{\mathrm{init}}$: 0.5, $\lambda_{\mathrm{sp}}$: 1.0 |
| | | ImageNet-1k | $lr$: 0.0001, $s_{\mathrm{init}}$: 0.5, $\lambda_{\mathrm{sp}}$: 1.0 |
| LwF | CIL/TIL | CIFAR-100 | $lr$: 0.01, $\alpha$: 0.5, $T$: 2.0, $wd$: 5e-5 |
| | | Tiny-ImageNet | $lr$: 0.01, $\alpha$: 0.5, $T$: 2.0, $wd$: 5e-5 |
| | | ImageNet-1k | $lr$: 0.001, $\alpha$: 0.5, $T$: 2.0, $wd$: 5e-5 |
| EWC | CIL/TIL | CIFAR-100 | $lr$: 0.01, $\lambda$: 10, $\gamma$: 1.0 |
| | | Tiny-ImageNet | $lr$: 0.01, $\lambda$: 10, $\gamma$: 1.0 |
| | | ImageNet-1k | $lr$: 0.001, $\lambda$: 25, $\gamma$: 1.0 |
| SI | CIL/TIL | CIFAR-100 | $lr$: 0.01, $c_{SI}$: 0.5, $\xi$: 1.0 |
| | | Tiny-ImageNet | $lr$: 0.01, $c_{SI}$: 0.5, $\xi$: 1.0 |
| | | ImageNet-1k | $lr$: 0.001, $c_{SI}$: 0.5, $\xi$: 1.0 |
| DyTox | CIL/TIL | CIFAR-100 | $lr$: 0.001 |
| | | Tiny-ImageNet | $lr$: 0.001 |
| | | ImageNet-1k | $lr$: 0.0001 |
| iCaRL | CIL/TIL | CIFAR-100 | $lr$: 0.01, $wd$: 5e-5 |
| | | Tiny-ImageNet | $lr$: 0.01, $wd$: 5e-5 |
| | | ImageNet-1k | $lr$: 0.001, $wd$: 5e-5 |
| DER++ | CIL/TIL | CIFAR-100 | $lr$: 0.01, $\alpha$: 0.2, $\beta$: 0.5 |
| | | Tiny-ImageNet | $lr$: 0.01, $\alpha$: 0.2, $\beta$: 1.0 |
| | | ImageNet-1k | $lr$: 0.001, $\alpha$: 0.2, $\beta$: 1.0 |

