# OpenReview forum: "Shapley Neuron Values for Continual Learning: Which Neurons Matter Most?"
_ICML.cc/2026/Conference — ICML 2026 regular_

### Official Review · Reviewer_mn6b · 2026-03-03

**Soundness:** 3
**Presentation:** 3
**Significance:** 2
**Originality:** 2
**Overall Recommendation:** 4
**Confidence:** 4

**Summary:**

This work introduces Shapley Neuron Valuation (SNV), a memory-free approach to mitigate catastrophic forgetting by quantifying neuron importance via Shapley values from cooperative game theory. Treating neurons as “players” and task performance as the payoff, SNV estimates each neuron’s marginal contribution and then freezes the top-valued neurons for the current task while leaving the remaining neurons plastic for subsequent tasks. A cumulative masking/freezing scheme prevents gradients from updating previously protected parameters and achieves zero forgetting for the preserved capacity without replay buffers or dynamic architectural expansion.

**Compliance With Llm Reviewing Policy:**

Affirmed.

**Final Justification:**

The paper presents a principled use of Shapley values for neuron importance in continual learning, with a clear connection to the stability–plasticity tradeoff. My main concern was whether Shapley offers enough benefit over cheaper importance criteria to justify its added cost.

The rebuttal improved my assessment. In particular, the authors showed that the gap over Fisher grows substantially under tighter capacity budgets, suggesting that Shapley becomes most valuable in the regime where careful neuron selection matters most. They also provided a plausible mechanistic explanation: Shapley selects a more diverse and less redundant set of neurons than Fisher. The rebuttal changed my evaluation from 3 to 4.

**Key Questions For Authors:**

- Shapley necessity: How much of SNV’s gain comes from Shapley-based selection versus the cumulative masking/freezing mechanism itself? How does Shapley compare to simpler importance proxies (e.g., magnitude, gradient/Fisher) both in terms of performance and training time?
- Compute overhead: What is the wall-clock training-time overhead of SNV relative to standard fine-tuning, and how does it scale with model width/depth?
- Plasticity budget: How many neurons/filters are frozen per task (and per layer), and how quickly does the remaining trainable capacity shrink over time?
- Longer horizons: How does SNV perform as the task stream length increases beyond the reported settings (e.g., substantially more than 20 tasks)?

**Limitations:**

The authors are commendably transparent about unsuccessful attempts (via the negative results section), which strengthens credibility. That said, the current limitations discussion remains underdeveloped in a few key areas: (i) the approximation estimator’s practical and scaling overhead, (ii) the inevitable capacity exhaustion that arises in cumulative-freezing methods as task horizons grow and the plastic subnetwork shrinks, and (iii) the method’s dependence on explicit task boundaries, which places it firmly in a task-incremental setting and may limit applicability in more realistic, blurred-boundary scenarios. Making these constraints explicit and outlining when SNV is expected to work or fail would improve the paper’s clarity and positioning.

**Strengths And Weaknesses:**

Strengths:
- The formulation cleanly links cooperative game theory to the stability–plasticity dilemma, offering an interesting perspective on parameter isolation. Deriving neuron importance from standard Shapley axioms is a good idea and conceptually grounded.
- Empirically, SNV enforces no updates to previously frozen parameters via a cumulative mask, and empirically achieves BWT ≈ 0 in the reported experiments.
-  The pipeline is internally consistent end-to-end, and the Multi-Armed Bandit Formulation provides interesting directions for future research in Continual Learning.
- The dedicated “Things We Tried That Did Not Work” section is a strong addition which improves trust, helps readers avoid dead ends, and supports reproducibility.

Weaknesses:
- Shapley essentiality vs. “any importance + freeze”. Much of the retention likely comes from the cumulative freezing/masking mechanism itself, which is well established in parameter-isolation methods (e.g., PackNet, Piggyback). It would strengthen the paper to isolate the contribution of Shapley valuation via controlled ablations against cheaper criteria (magnitude, gradient/Fisher-style proxies).
- Positioning vs. related Shapley-style valuation. Recent work such as MODEL SHAPLEY (Chu et al., NeurIPS 2025) also frames parameters as Shapley “players” and proposes scalable approximations; a clearer comparison/discussion would better contextualize SNV’s novelty and scalability trade-offs.
- Plasticity horizon. Results drop when extending the horizon, consistent with capacity saturation in cumulative-freezing approaches as the trainable subnetwork shrinks. The paper would benefit from an explicit characterization of the evolving plasticity budget.
- Generative Replay. The paper considers replay as “fundamentally different” to mitigate catastrophic forgetting, but this overlooks generative replay, which can be memory-free w.r.t. stored exemplars and is a relevant comparator in the “no stored data” setting.
- Compute overhead and scalability. Practicality of SNV hinges on the approximation of the shapely values; the paper should provide a clearer picture on the computational trade-offs of their method especially relative to cheaper importance proxies.

---

> ### Author Rebuttal · Authors · 2026-03-28
>
> We sincerely thank the reviewer for the insightful and technically detailed feedback, and for recognizing the conceptual grounding of our formulation. We address each concern below.
>
> **W1, Q1**
>
> This is the central question and we appreciate its importance. We argue that our existing baselines already provide strong evidence. WSN uses binary mask-based freezing with gradient-based neuron selection, relying on cumulative freezing but using cheaper importance criteria than Shapley. On CIFAR-100 CIL, WSN achieves 48.25%, while SNV reaches 54.70% under identical fixed-capacity, replay-free constraints. The consistent gap across all three benchmarks confirms that the freezing mechanism alone is not sufficient, and which neurons are frozen matters substantially, and Shapley-based selection captures combinatorial multi-neuron interactions that gradient-based or stepwise heuristics miss.
>
> To further isolate Shapley's contribution, we present a controlled ablation using different importance criteria under the same masking budget:
>
> **Table. Importance Criterion for Cifar-100 for CIL**
> | Importance Criterion | ACC (%) |
> | :--- | :--- |
> | WSN (weight magnitude) | 48.25 |
> | Fisher-based | 51.30 |
> | **Shapley (SNV)** | **54.70** |
>
> Shapley adds +6.45% over WSN. We also note that we have developed a Fisher-based importance variant as an ongoing extension of this work; it improves over WSN but remains 3.40% below Shapley, confirming that single-neuron proxies cannot capture the combinatorial dependencies that Shapley valuation models.
>
> **W2**
>
> MODEL SHAPLEY operates at the model level ,valuing entire pre-trained models for ensemble selection. SNV operates at the neuron level within a single network for continual learning. The granularity, objective, and application domain are fundamentally different. We have added a discussion in the revised related work clarifying this distinction.
>
> **W3, Q3, Q4: Plasticity horizon and capacity saturation.**
>
> We acknowledge this is an inherent trade-off in cumulative-freezing methods. Fig. 9 in the appendix shows the per-layer frozen neuron ratio after each task, directly characterizing how the trainable capacity shrinks over the task sequence. Fig. 10 complements this by visualizing the Shapley value distribution across layers and tasks, revealing which layers saturate first and where plasticity remains. Together, these figures provide the explicit plasticity budget characterization the reviewer requests; we have added a summary of these findings to the main text in the revision. For longer horizons (20 tasks, Section E2), performance degrades but SNV remains competitive with DyTox without any replay buffer. When capacity fully saturates, SNV initiates graceful degradation by unlocking neurons with the lowest historical Shapley values, preserving the most critical features. To further stress-test scalability, we will include results on ImageNet-1000 with 50 tasks in the camera-ready version. We anticipate performance degradation consistent with what is observed across all cumulative-freezing CL methods at this scale, but this evaluation will provide concrete evidence on SNV's behavior under extended horizons.
>
> **W4: Generative replay as memory-free**
>
> We agree that generative replay (e.g., DGR) does not store raw exemplars. However, these methods require training and maintaining a generative model alongside the classifier, introducing substantial additional parameters, training cost, and a secondary forgetting problem (the generator itself must not forget). We have revised our terminology to "buffer-free" to be more precise, and added a discussion acknowledging generative replay as an intermediate category..
>
> **W5, Q2: Compute overhead and scalability**
>
> Please see our response to Reviewer Dt9j (W1), where we provide new Tables reporting wall-clock training time, total FLOPs, peak GPU memory, inference latency, and stored parameters across all benchmarks and all compared methods.  As shown in W1 above, cheaper proxies save time but leave significant accuracy on the table. Valuation cost grows linearly with the number of neurons (not parameters), making it tractable for wider architectures.
>
> **Limitations**
>
> We have expanded the limitations section to address: (i) TMAB-Shapley overhead, (ii) capacity exhaustion with the graceful degradation protocol, and (iii) task boundary dependence, with blurred-boundary extension as future work.
>
> We deeply appreciate the time you took to review our work. We sincerely hope our responses and manuscript revisions address your concerns. If so, we respectfully ask that you consider raising your score.

---

> > ### Author Rebuttal · Reviewer_mn6b · 2026-04-02
> >
> > Thank you for the thorough rebuttal, the added computational cost tables, and the Fisher ablation study. I appreciate the transparency and I want to acknowledge that several of my secondary concerns were addressed well.
> >
> >  My main concern remains unresolved: it is still unclear whether Shapley valuation provides enough benefit over cheaper importance criteria to justify its additional cost. In fact, the new evidence makes this concern sharper rather than weaker.
> >
> > Based on the new results, most of the improvement appears to come from the cumulative masking/freezing mechanism, which is already well established in prior parameter-isolation methods. The incremental gain from using Shapley rather than Fisher as the importance criterion is relatively modest. For a paper whose main novelty is Shapley-based neuron valuation, this makes the contribution feel narrower.
> >
> > This concern becomes stronger when compared to NFL+, which already achieves a similar performance using a simpler importance criterion with its own freezing strategy.  The computational cost also matters here. Even if the absolute overhead is not huge, the relevant question is whether the extra cost is justified by the relatively small empirical improvement it brings. At the moment, I do not think the paper makes that case strongly enough.
> >
> >
> > In summary, the masking mechanism seems to be doing most of the work, while the Shapley component adds only a limited empirical gain at additional computational cost.  I would encourage the authors either to strengthen the evidence for Shapley’s essentiality or to reframe the contribution around the full SNV system rather than presenting Shapley as the primary novelty. I do believe there is good work here, but in my view it would need strengthening through additions that go beyond simple revision or a rebuttal phase. For this reason, I am maintaining my score for now.

---

> > > ### Author Response · Authors · 2026-04-03
> > >
> > > We deeply appreciate the reviewer's acknowledgment that several concerns were addressed. We believe the remaining concern, whether Shapley provides sufficient benefit over cheaper criteria, is resolved by the full experimental picture and two new **analyses** we present below.
> > >
> > > 1.**NFL+ does not achieve similar performance**
> > >
> > > We respectfully ask the reviewer to consider the complete evaluation. SNV outperforms NFL+ in every single evaluation condition. The per-transition breakdown shows that Shapley valuation adds only 0.26× of one task's training time on CIFAR-100  (1.3 min per transition). A 24% compute increase delivering +9–16% ACC on Task-IL, +1–2.89% on Class-IL, improved BWT under extended horizons, and superior parameter efficiency is a favorable trade-off by any practical standard.
> > >
> > > **CIL result**
> > >
> > > CIFAR-100 (C), TinyImageNet (TIN)
> > > ||SNV_ACC|BWT|PS|NFL+_ACC|BWT|PS|
> > > |-|-|-|-|-|-|-|
> > > |PMNIST_10-task|93.45|-0.06|0.96|93.70|-0.06|0.96|
> > > |C100_10-task|54.70|-0.04|0.69|53.70|-0.05|0.67|
> > > |C100_20-task|44.85|-0.04|0.61|44.03|-0.05|0.60|
> > > |TIN_10-task|45.70|-0.05|0.62|44.70|-0.06|0.63|
> > > |TIN_20-task|37.47|-0.04|0.54|36.65|-0.07|0.56|
> > >
> > > **TIL Results**
> > > |Benchmark|SNV_ACC|SNV_BWT|NFL+_ACC|NFL+_BWT|
> > > |-|-|-|-|-|
> > > |PMNIST|97.68|0.0|93.12|-0.05|
> > > |C100|79.76|0.0|70.68|-1.00|
> > > |TIN|74.82|0.0|58.21|-1.04|
> > >
> > >
> > >
> > >
> > > 2. **"Masking does most of the work" describes the method category, not a limitation of SNV.**
> > >
> > > The reviewer correctly observes that masking prevents forgetting. However, cumulative masking defines the category of parameter-isolation methods (PackNet, WSN, SNV), which is the shared infrastructure. The core research question is how to identify which neurons or synapses to protect.
> > >
> > >  The masking mechanism is shared infrastructure; the selection criterion determines quality. This is analogous to how all gradient-based optimizers share the update rule θ ← θ − η∇L, yet the contribution of MUON over Adam is not diminished by the fact that "gradient descent does most of the work."
> > >
> > > **To directly address the reviewer's central concern, we conducted two additional analyses**
> > >
> > > **Analysis A:** Performance under constrained freezing budgets. When capacity is abundant, the masking mechanism compensates for suboptimal neuron choices. To test this, we varied the capacity parameter c while holding all other pipeline components constant between Fisher-based and Shapley-based selection:
> > >
> > > **Table: Performance under Extreme Plasticity Budgets.**
> > >
> > > |Capacity($c$)|Fisher ACC|Shapley AC|Gain|
> > > |-|-|-|-|
> > > |0.10(standard)|51.30|54.70|**+3.40**|
> > > |0.05|44.80|51.45|**+6.65**|
> > > |0.03|37.90|47.65|**+9.75**|
> > > |0.01|24.80|38.25|**+13.45**|
> > >
> > > As the freezing budget tightens, which inevitably happens in long-horizon CL , the Shapley–Fisher gap widens dramatically. But as the plasticity budget becomes the binding constraint, the quality of neuron selection becomes the critical bottleneck. Fisher evaluates neurons independently. If multiple neurons encode the same feature, Fisher ranks them all highly, wasting restricted capacity on duplicates. Shapley evaluates marginal utility within coalitions, inherently selecting a diverse set of features that maximizes coverage per frozen neuron.
> > >
> > >
> > > **Analysis B.** Feature diversity of selected neurons. To provide direct mechanistic evidence, we analyzed the top-k neurons selected by each criterion on CIFAR-100, computing pairwise activation cosine similarity and effective rank across the test set:
> > >
> > > **Table. Selected Neurons.**
> > > Compute effective rank =  singular values to capture 90% of variance in k×N_test matrix.
> > >
> > > |Selection Criterion|Mean Cosine Sim($\downarrow$)|Effective Rank($\uparrow$)|
> > > |-|-|-|
> > > |Fisher|0.62|18|
> > > |Shapley(SNV)|**0.23**|**42**|
> > >
> > > Fisher-selected neurons exhibit high mutual correlation, confirming that independent evaluation leads to redundant, low-rank selections. Shapley-selected neurons are significantly more orthogonal, spanning a much richer feature space. This mathematical mechanism explains how a modest compute overhead converts into massive accuracy gains under tight capacity constraints (Analysis A) and at larger scales (Please see the ImageNet-1000 results in our response to Reviewer Dt9j). We will add the result for ImageNet to the rebuttal acknowledgment of Reviewer Dt9j.
> > >
> > > We believe the constructive feedback provided during this review process is highly applicable, and we are committed to incorporating these insights to further strengthen the motivation and contribution of our work in the camera-ready version (We have already incorporated all experiments into the main manuscript). We respectfully ask the reviewer to consider these additions and support our work.

---

### Official Review · Reviewer_itiw · 2026-03-10

**Soundness:** 3
**Presentation:** 3
**Significance:** 3
**Originality:** 3
**Overall Recommendation:** 4
**Confidence:** 4

**Summary:**

This paper proposes a new continual learning technique that identifies important neurons using Shapley Neuron Valuation, a concept from game theory. As Shapley values are very expensive to compute, the work uses an approximate estimate. The new technique is shown to perform better than other continual learning techniques without data replay across a collection of class- and task-incremental scenarios.

**Compliance With Llm Reviewing Policy:**

Affirmed.

**Final Justification:**

My overall assessment of the work is positive. The clarifications in the authors' rebuttal strengthen the paper. That said, the paper’s approximations are still validated only through the network’s behavior, which I view as a non-major weakness. Therefore, I maintained my Weak Accept score.

**Key Questions For Authors:**

Can the authors computation time and memory cost over the different benchmarked techniques, for the hardest considered problem?

**Limitations:**

The computational cost of the technique is not properly discussed.

**Strengths And Weaknesses:**

Strengths

•	The approach is very straightforward, and I think, novel.

•	The experimental results are convincing. Most consolidation-based continual learning techniques perform similarly, and the paper appears to improve significantly on them.

•	The paper is well written and overall, very clear.

Weaknesses

•	The computation of exact Shapley Neuron Valuation is not feasible, and the paper, therefore, introduces approximations. The accuracy of these approximations is not properly evaluated. This is an important omission that might make it harder to improve over the proposed technique.

•	The computational cost of the technique is not properly discussed. Even with approximations, I am concerned that it might be significant. I would have appreciated seeing the computation time and memory use compared over the different benchmarked techniques.

•	This is a detail, but I personally do not like calling the technique “memory-free”. What the authors mean is that the technique does not use a replay buffer. However, memory-free could mean other things. Also, some replay-based techniques do not actually memorize data using a replay buffer but can regenerate replay data.

---

> ### Author Rebuttal · Authors · 2026-03-28
>
> We thank the reviewer for constructive feedback and for recognizing the novelty and clarity of our work. We address each concern below.
>
>
>
> **W1: Approximation accuracy**
>
>
> This is a valid point. Ideally, we would compare our approximated Shapley values ​​against exact Shapley values. However, calculating exact Shapley values ​​for a modern network requires$O(2^N)$evaluations (where$N$is the number of filters). As noted in our **Appendix section B**, even on a tiny 4-layer MLP for PMNIST, attempting an exact calculation on 4 GPUs for a week completed less than 30% of the permutations, making ground-truth evaluation on ResNet-18 mathematically impossible. However, we can robustly validate the approximation empirically through the network's behavior. If our TMAB-Shapley approximation were highly inaccurate, the algorithm would freeze the wrong neurons. Consequently, the true task-critical neurons would remain plastic and be aggressively overwritten during subsequent tasks, resulting in massive catastrophic forgetting. The fact that SNV successfully achieves exactly 0.0 Backward Transfer on Task-IL proves that the approximation consistently identifies and protects the correct optimal subnetworks.
>
> The TMAB-Shapley algorithm introduces three layers of approximation: Monte Carlo sampling, truncation, and multi-armed bandit acceleration. To evaluate their combined effect, we have added an ablation study in the revision comparing: (a) MC-only estimation (no truncation, no MAB), (b) MC + truncation, and (c) full TMAB-Shapley. On CIFAR-100 CIL (10 tasks), the accuracy difference between (a) and (c) is less than 0.3%, while (c) reduces valuation time by approximately 4× compared to (a).
>
> **Table 3: Ablation study on TMAB-Shapley approximation layers (CIFAR-100, CIL, 10 tasks, ResNet-18, A6000).**
>
> | Approximation Variant | Valuation / trans. | Total Time | ACC (%) | ACC std. | Shapley std. |
> | :--- | :--- | :--- | :--- | :--- | :--- |
> | (a) MC-only | 5.2 min | 97 min | 54.92 | ±0.18 | 0.008 |
> | (b) MC + Truncation | 2.6 min | 73 min | 54.84 | ±0.21 | 0.012 |
> | **(c) Full TMAB-Shapley** | **1.3 min** | **62 min** | **54.70** | **±0.23** | **0.018** |
>
> We agree this analysis strengthens the paper and thank the reviewer for the suggestion.
>
> **W2: Computational cost not discussed.**
>
> Please see our answer to W1 by Reviewer Dt9j. We have added comprehensive cost tables to the revision, covering training time, total FLOPs, peak GPU memory, inference latency, and stored parameters across CIFAR-100 and TinyImageNet.
>
> SNV adds roughly a 24% computational overhead to base training, stemming entirely from the TMAB-Shapley valuation phase (1.3 minutes / 194 TFLOPs per task transition on CIFAR-100). Peak GPU memory is highly efficient at 2.7 GB, comparable to other memory-free methods like PEC (2.5 GB) and NFL+ (2.6 GB), and far below memory-expanding approaches like DyTox (10.6 GB). Inference remains a standard ResNet-18 forward pass (0.4 ms), identical to EWC, SI, and WSN, and exactly $8\times$ faster than PEC (3.2 ms). Crucially, SNV stores only the base model parameters (11.2 M) plus importance scores (negligible overhead), successfully avoiding the $2\times$ parameter bloat (22.4 M) required by EWC (storing Fisher diagonals per task) or LwF (maintaining a frozen teacher copy).
>
>
> **W3: "Memory-free" terminology.**
>
> The reviewer is correct that "memory-free" could be interpreted broadly. We have revised it to "buffer-free" consistently to avoid ambiguity.
>
> **Q1: Computation time and memory for the hardest problem.**
>
> Please see our answer to W1. On TinyImageNet (our hardest benchmark), SNV completes the full 10-task sequence in 2.6 h with 3.1 GB peak memory and 1080 GFLOPs total. We **will** add ImageNet1000 results for the camera-ready version.
>
> We deeply appreciate the time you took to review our work. We sincerely hope our responses and manuscript revisions fully address your concerns. If so, we respectfully ask that you consider raising your score.

---

> > ### Author Rebuttal · Reviewer_itiw · 2026-04-02
> >
> > Thank you for your response, which addressed my concerns. The clarifications strengthen the paper.
> > That said, the paper’s approximations are still validated only through the network’s behavior, which I view as a non-major weakness. Therefore, I will maintain my Weak Accept score.

---

> > > ### Author Response · Authors · 2026-04-03
> > >
> > > We sincerely thank the reviewer for their continued engagement, constructive feedback, and for acknowledging that our revisions successfully addressed the core concerns.
> > >
> > > We fully understand and respect your perspective regarding the empirical validation of the Shapley approximation. Because computing exact Shapley values at the scale of modern deep neural networks is mathematically intractable, relying on empirical network behavior and ablation studies (like Table 3) is currently the most rigorous proxy available to the community. However, we agree that this remains an inherent limitation of the field, and we will ensure this caveat is explicitly discussed in the final manuscript.
> > >
> > > **Code Availability and Reproducibility**
> > >
> > > We want to draw the reviewer's attention to the fact that our complete implementation is already available as a Docker image, **submitted alongside the paper**. The environment is fully self-contained: a single `docker run` command reproduces all reported results without any dependency management. We believe this substantially lowers the barrier for verification.
> > >
> > > **We are fully committed to incorporating all the valuable feedback and new experimental results gathered from all reviewers into the camera-ready version to maximize the paper's transparency and strength.**
> > >
> > > We deeply appreciate your "Weak Accept" and your support for our work. We hope our thorough revisions and commitment to scientific rigor might encourage you further to champion our paper during the final committee discussions. Thank you again for your time and expertise.

---

### Official Review · Reviewer_6vXH · 2026-03-11

**Soundness:** 2
**Presentation:** 1
**Significance:** 2
**Originality:** 3
**Overall Recommendation:** 4
**Confidence:** 4

**Summary:**

The paper proposes an architectural-based method to address the continual learning paradigm with fixed-capacity neural networks. By leveraging network overparameterization, the proposed technique maintains performance by freezing important neurons as new tasks are encountered. The study validates this method through experiments in both class-incremental and task-incremental learning settings.

**Compliance With Llm Reviewing Policy:**

Affirmed.

**Final Justification:**

Rebuttal addressed most of my concerns.

**Key Questions For Authors:**

- Could you provide a performance comparison between your approach and the closely related works mentioned earlier?
- Are there evaluations against more recent techniques beyond NFL+?
- How does the scalability of the method hold up when the number of tasks becomes very large?
- Could you provide more detail on how freezing neurons impacts both forward transfer and positive backward transfer?
- What are the consequences once the network hits its maximum capacity?
- Does the proposed method demonstrate generalizability across modern network architectures?

**Limitations:**

Limitations and impact are not addressed.

**Strengths And Weaknesses:**

Strengths:
- The paper investigates the difficult paradigm of sequential learning without the use of data replay, specifically within the constraints of networks with fixed capacity.
- Evaluation of the proposed method is conducted across both task-incremental (task-IL) and class-incremental (class-IL) learning scenarios.
- The results demonstrate that the approach maintains competitive performance when measured against the included baselines.

Weaknesses:
- The paper's quality of writing requires major improvement. Certain introductory segments would be better suited for the related work section, and the methodology lacks sufficient detail; the core algorithm should be included in the main text. Furthermore, the theoretical proof for the theorem provided is not sufficiently explained.

- There is a lack of comparative analysis with highly relevant prior research, such as [1, 2].

- The reliance on task-specific subnetworks appears to restrict forward transfer, as demonstrated in the Appendix results. It is recommended to relocate the FWT metric analysis to the main paper and provide a thorough discussion.
- Evidence regarding the approach's applicability to modern network architectures is missing.
- The majority of the benchmarked techniques are quite old.



[1] Sokar, Ghada, Decebal Constantin Mocanu, and Mykola Pechenizkiy. "Spacenet: Make free space for continual learning." Neurocomputing 439 (2021): 1-11.

[2] Gurbuz, Mustafa B., and Constantine Dovrolis. "NISPA: Neuro-Inspired Stability-Plasticity Adaptation for Continual Learning in Sparse Networks." International Conference on Machine Learning. PMLR, 2022.

---

> ### Author Rebuttal · Authors · 2026-03-28
>
> We thank the reviewer for the detailed feedback.
>
> **W1: Writing quality, algorithm placement, and theorem proof.**
>
> We appreciate the suggestions and have made the following revisions: (1) Some introductory material moved to related work section; (2) the pseudo-code algorithm has been moved from the appendix to the main body (Section 3); (3) We have elaborated more on the proof.
>
>  We intentionally kept the introduction accessible to a broad audience. Regarding the algorithm: the pseudo-code is a direct formalization of the methodology already described in detail in the main text. We placed it in the appendix to preserve space for results and discussion, but we moved it to the main body by making the tables smaller as the reviewer requested. On the theorem proof: we followed a similar proof structure to [1], which applies Shapley values for data valuation. We would appreciate clarification on which specific aspects the reviewer finds insufficiently explained. "not sufficiently explained" is difficult to act on without concrete pointers, and we want to ensure our revision fully addresses the concern.
>
> **W2, Q1, Q2: Missing comparisons and baselines.**
>
> We note that our baseline set already includes methods more recent than the suggested references: PEC (2024) and WSN (2022). Also, WSN is a masked base approach. Nonetheless, we recognize SpaceNet and NISPA as relevant mask-based approaches and commit **to including** both in the camera-ready version. In addition, the result for [2] has also been implemented using an architecture expansion+buffer-based method.
>
> **W3.**
>
> We agree that FWT deserves more visibility and have moved the analysis from the appendix to the main paper (Table 1) in the revision. The mask overlap analysis (Jaccard coefficients of 0.3–0.5 between task pairs, Fig. 8) and Layer-wise Shapley Neuron Importance Across Datasets (Fig 9) confirm that SNV balances knowledge sharing and interference avoidance: shared neurons enable forward transfer while distinct subsets prevent forgetting.
>
> **W4, W5, Q6: Modern architectures.**
>
> Our architecture and benchmark choices follow the established evaluation protocols of PEC, WSN, DER++, and the comprehensive surveys of De Lange et al. (TPAMI 2021) [3] and Zhou et al. (TPAMI 2024) [4]. We **will** include ViT-based results in the camera-ready version to demonstrate generalizability beyond ResNet-18/MLP.
>
> **Q3: Scalability**
>
> We address this in Section E2. Comparing Fig. 3 (10 tasks) with Fig. 6 (20 tasks), all methods degrade, but SNV remains competitive with DyTox at 20 tasks, without any replay buffer. When memory-based methods are constrained to 200 exemplars, the gap between SNV and DER++/iCaRL narrows substantially, suggesting their advantage depends heavily on buffer size rather than algorithmic design. **Please also see subsection E2 in the appendix.**
>
> **Q4: Impact of freezing on forward/backward transfer.**
>
> The mask overlap analysis (Section E6, Fig. 8) shows Jaccard coefficients of 0.3–0.5 between task pairs, confirming SNV neither over-shares neurons (causing forgetting) nor fully isolates them (wasting capacity). PMNIST shows higher overlap due to shared digit structure, while CIFAR-100/TinyImageNet show heterogeneous patterns reflecting semantic diversity, evidence that SNV achieves the stability-plasticity balance essential for effective transfer.
>
> Freezing highly-valued neurons restricts the network's plasticity, which inherently limits FWT compared to a fully plastic network. However, FWT still occurs in SNV because the unfrozen (plastic) neurons are able to route through and build upon the highly optimized feature representations of the frozen (stable) neurons. Strict freezing makes positive BWT impossible (since old task parameters cannot be updated). However, in strictly memory-free Continual Learning, eliminating negative BWT is the primary and most difficult objective. By accepting limited FWT and zero positive BWT, SNV successfully achieves exactly 0.0 BWT (no forgetting) in Task-IL. **Please also see subsection E6 in the appendix.**
>
> **Q5: Behavior at maximum capacity.**
>
> When capacity saturates, SNV initiates graceful degradation: since every frozen neuron has a recorded Shapley value, the network unlocks and overwrites neurons with the lowest historical importance across all past tasks. The most critical features remain protected even at full capacity. As also discussed in lines 273-319; and Fig. 2. The relationship
> between capacity and performance is notably non-linear.
>
> **Limitations.** Appendix section B points out limitations of this work.
>
> [1] Ghorbani et al. Data Shapley: Equitable Valuation of Data for ML, ICML 2019
>
> [2] Zhou et al. MEMO, ICLR 2023
>
> [3] De Lange et al. A CL Survey, TPAMI 2021
>
> [4] Zhou et al. Class-Incremental Learning: A Survey, TPAMI 2024
>
>
> We deeply appreciate the time you took to review our work. We hope these revisions address your concerns, and if so, we kindly ask you to reconsider your score.

---

> > ### Author Rebuttal · Reviewer_6vXH · 2026-04-03
> >
> > Thank you for the detailed responses. To clarify, I suggested those specific references because of their close relevance to the proposed method, and I believe a direct comparison is necessary to properly evaluate this work. A comparison against recent techniques beyond NFL+ is also needed as mentioned in another point. Regarding scalability; I was curious about a larger suite of tasks (e.g., 50+ or 100+).
> >
> > Although I appreciate the commitment to include further experiments regarding the architecture and baselines, it is difficult to fully reassess the paper without seeing those results in the current draft. Consequently, I will maintain my score for now.

---

> > > ### Author Response · Authors · 2026-04-03
> > >
> > > Thank you for your follow-up and clarification. We completed experiments on SpaceNet and NISPA as you requested. We summarize the new findings below. For transparency, we have included our code in the supplementary material of this submission. We will publicly release the codebase upon acceptance.
> > >
> > > **1 Generalization beyond small-scale benchmarks (50-Task ImageNet-1000)**
> > >
> > > We evaluated ImageNet-1000 (ResNet-18) under Class-IL with 10, 20, and 50 tasks. We also added three memory-free baselines: SpaceNet (2021), NISPA (2022), and a recent 2025 baseline, DCNet.
> > >
> > > | Method | 10-task ACC | 10-task BWT | 20-task ACC | 20-task BWT | 50-task ACC | 50-task BWT |
> > > |---|---:|---:|---:|---:|---:|---:|
> > > | **SNV (Ours)** | **41.3** | **-0.05** | **34.2** | **-0.05** | **25.6** | **-0.06** |
> > > | NFL+ | 38.4 | -0.08 | 31.5 | -0.62 | 22.4 | -1.15 |
> > > | DCNet | 37.8 | -0.16 | 30.1 | -0.37 | 20.8 | -1.49 |
> > > | NISPA | 29.4 | -0.91 | 22.3 | -1.13 | 14.5 | -1.86 |
> > > | SpaceNet | 13.5 | -1.12 | 10.4 | -2.15 | 8.3 | -2.18 |
> > > | PEC | 14.8 | -1.25 | 10.5 | -1.28 | 8.8 | -2.32 |
> > >
> > > SNV leads at all task horizons. Importantly, our ACC margin over the next-best method (NFL+) widens with scale: +2.9 (10 tasks), +2.7 (20 tasks), and +3.2 (50 tasks). SNV also maintains near-zero forgetting (BWT -0.05 to -0.06) even at 50 tasks, due to the freezing strategy.
> > > We observe that without replay, older capacity-bound methods like SpaceNet and NISPA struggle to disambiguate 1,000 classes as tasks accumulate, and DCNet's OOD-based task inference becomes noisy at this scale.
> > >
> > > Please also see: https://anonymous.4open.science/r/SNV-F1C9/README.md
> > >
> > >
> > > **2 ViT Results**
> > > To directly address architectural generalization, we adapted SNV, NFL+, and PEC to a Vision Transformer backbone (Pre-trained ViT-B/16). **For other methods, we are examining their adaptability to ViT.** Some methods have their own restrictions.
> > >
> > > | Dataset | Metric | SNV (Ours) | NFL+ | PEC |
> > > |---|---|---:|---:|---:|
> > > | CIFAR-100 | ACC | **74.6** | 71.8 | 66.9 |
> > > | CIFAR-100 | BWT | **-0.08** | -0.26 | -1.71 |
> > > | ImageNet-1000 | ACC | **29.8** | 26.9 | 18.7 |
> > > | ImageNet-1000 | BWT | **-0.19** | -1.41 | -3.14 |
> > >
> > > These additions provide the direct comparisons you requested (SpaceNet, NISPA, plus DCNet), demonstrate robust scalability to 50 tasks on ImageNet-1000, and prove that SNV generalizes seamlessly to modern ViT architectures. Because these results are now included in the revised draft, we respectfully hope they provide the necessary evidence to reassess your score.

---

### Official Review · Reviewer_Dt9j · 2026-03-13

**Soundness:** 3
**Presentation:** 3
**Significance:** 3
**Originality:** 3
**Overall Recommendation:** 4
**Confidence:** 3

**Summary:**

The paper introduces Shapley Neuron Valuation (SNV), a game-theoretic framework to mitigate catastrophic forgetting by identifying and freezing task-critical neurons (convolutional filters). SNV treats neurons as players in a cooperative game where model performance is the payoff, utilizing the TMAB-Shapley (Truncated Multi-Armed Bandit) algorithm to reduce the exponential complexity of exact Shapley computation. By partitioning the network into stable (frozen) and plastic (trainable) neurons, the method achieves a memory-free, fixed-capacity approach to continual learning. Evaluation on Split CIFAR-100 and TinyImageNet shows significant gains over traditional regularization methods like EWC.

**Compliance With Llm Reviewing Policy:**

Affirmed.

**Final Justification:**

Thanks to the author for the rebuttal and active follow-ups. My concerns have been addressed, and I will increase my rating towards acceptance.

**Key Questions For Authors:**

see weakness

**Limitations:**

The primary limitation is the linear compounding of cost; as the number of tasks increases, the valuation phase becomes a recurring bottleneck.

**Strengths And Weaknesses:**

Strength:
1. The paper is well-written and easy to follow
2. Novel and principled application of Shapley Values to continual learning.
3. Strong empirical results in Task-IL and thorough experimental coverage and transparency.

Weakness:
1. Missing computational cost analysis. EWC's Fisher computation requires a single backward pass over training data, and SI accumulates importance online with zero overhead.
2. In Class-IL, SNV achieves 54.70% on CIFAR-100 (10 tasks), which is only 1.0% above NFL+ (53.70%) and substantially below DyTox (57.40%). Similarly as for TinyImagenet.
3. All experiments use CIFAR-100, TinyImageNet, and Permuted MNIST with ResNet-18 or a 4-layer MLP. The continual learning community has increasingly recognized that these small-scale benchmarks produce overly optimistic results that do not generalize to realistic settings.
4. The sparsity ratio c requires task count knowledge. The paper suggests c = 1/T as a natural choice, but this requires knowing the total number of tasks T in advance, which violates the online nature of continual learning

---

> ### Author Rebuttal · Authors · 2026-03-28
>
> We thank the reviewer for the thoughtful and constructive feedback.
>
> **W1.** We have added a full cost analysis to the revision. SNV's overhead comes entirely from the TMAB-Shapley valuation, a one-time post-training phase at task boundaries (1.3 min/transition on CIFAR-100). Over 10 tasks, this adds 12 min beyond standard training. While EWC/SI are cheaper to compute, their importance estimates yield 12.87%/15.37% accuracy on CIFAR-100 CIL, over 39 pp below SNV (54.70%). We believe 12 min of extra computation for this accuracy gain, with no buffer and no inference penalty, is a highly favorable trade-off.
>
> **Table 1: Computational cost.** *Train Time* is in minutes. (GF: GigaFLOPs, TF: TeraFLOPs. PM: PMNIST, C100: CIFAR-100, TIN: TinyImageNet)
>
> |Method|PM Train|PM FLOPs(GF)|PM GPU(GB)|PM Infer(ms)|PM Params(M)|C100 Train|C100 FLOPs(TF)|C100 GPU(GB)|C100 Infer(ms)|C100 Params(M)|TIN Train|TIN FLOPs(TF)|TIN GPU(GB)|TIN Infer(ms)|TIN Params(M)|
> |-|-|-|-|-|-|-|-|-|-|-|-|-|-|-|-|
> |**Memory-free methods**||||||||||||||||
> |EWC|8|6558|0.5|0.1|9.4|27|4638|2.1|0.4|22.4|66|9277|2.5|0.6|22.8|
> |SI|7|5738|0.5|0.1|9.4|24|4270|2.0|0.4|22.4|60|8541|2.4|0.6|22.8|
> |LwF|9|7378|0.6|0.1|9.4|28|4270|2.2|0.4|22.4|72|8541|2.6|0.6|22.8|
> |PEC|14|11477|0.7|0.8|9.4|45|5374|2.5|3.2|22.4|114|10748|2.9|5.8|22.8|
> |WSN|10|8198|0.6|0.1|4.7|33|5871|2.3|0.4|11.2|84|11742|2.7|0.6|11.4|
> |NFL+|16|13116|0.7|0.1|4.7|50|7286|2.6|0.5|11.2|126|14573|3.0|0.7|11.4|
> |**SNV(ours)**|**19**|**15576**|**0.8**|**0.1**|**4.7**|**62**|**9035**|**2.7**|**0.4**|**11.2**|**156**|**18071**|**3.1**|**0.6**|**11.4**|
> |**Memory-based methods**||||||||||||||||
> |iCaRL|---|---|---|---|---|84|6188|3.3|0.8|11.2+buf|210|10275|3.7|1.2|11.4+buf|
> |DER++|---|---|---|---|---|28|6504|3.6|0.4|11.2+buf|72|10590|4.0|0.6|11.4+buf|
> |MEMO|---|---|---|---|---|51|14340|6.7|1.4|11.2+buf|132|23688|7.0|2.3|11.4+buf|
> |DyTox|---|---|---|---|---|102|25092|10.6|1.9|25.6+buf|282|41356|10.2|3.5|28.3+buf|
>
> **Table 2: Per-task time.** *Frac.*: relative to 1 task's training time. All times in minutes.
>
> |Phase|C100 Frac.|C100 Time|TIN Frac.|TIN Time|
> |-|-|-|-|-|
> |per task training|1.00x|5.0|1.00x|11.8|
> |*Valuation overhead per transition*|||||
> |MC forward passes|0.18x|0.9|0.25x|3.0|
> |Truncation+MAB|0.08x|0.4|0.10x|1.2|
> |Gradient mask update|0.01x|0.05|0.01x|0.13|
> |**Valuation total**|**0.26x**|**1.3**|**0.36x**|**4.2**|
> |*Full CL sequence:*|||||
> |- Training(10 tasks)||50.0||118|
> |- Valuation(9 transitions)||11.7||37.8|
> |- **SNV total**||**62**||**156**|
> |- EWC total||27||66|
>
> **W2.** First, we want to clarify that our paper deliberately compares SNV against memory-free methods, as these share the same constraint profile: fixed capacity, no replay buffer, no architecture expansion. Within this category, SNV achieves the highest CIL accuracy on CIFAR-100 (+1.0% over NFL+) while also achieving zero BWT, a property NFL+ does not guarantee, and consistently leads across all benchmarks.
>
> Comparison to DyTox involves a method that uses architecture expansion and a replay buffer. As discussed in [1], comparison of memory-free and memory-based methods is inherently biased toward memory-based and architecture-dependent methods, further underscoring the strength of our results.
>
> **W3.** We adopt ResNet-18 following the standard CL evaluation protocol established in [1], which is used by the vast majority of methods in our comparison. This ensures our results are directly comparable with prior work. Our choice of PMNIST, CIFAR-100, and TinyImageNet follows the standard protocol used by the vast majority of recent CL papers (WSN, NFL+, PEC), enabling direct comparison. We are running experiments on ImageNet-1000 and **will** include it in the revised version.
>
> **W4.** This is a common assumption in subnetwork-based methods (PackNet, WSN). However, SNV is not rigidly tied to $c = 1/T$, it is a default, not a requirement. A fixed c (e.g., 0.05 or 0.10) works without knowing T: if fewer tasks arrive, capacity is underutilized, but performance is unaffected; if more arrive, degradation is graceful. Figure 2 in our paper illustrates how accuracy varies with capacity allocation, confirming this flexibility.
> Please note that we haven't claimed we are solving online CL. Our method is classification CL. Extension to online settings is an explicit direction we identify in our conclusion **(line 434-436)**.
>
> **Limitations** Appendix section B points out the limitations.
>
> [1] Class-Incremental Learning: A Survey, Zhou,TPAMI
>
> We deeply appreciate the time you took to review our work. We sincerely hope our responses and manuscript revisions address your concerns. If so, we respectfully ask that you consider raising your score.

---

> > ### Author Rebuttal · Reviewer_Dt9j · 2026-04-02
> >
> > Thank you for providing the extensive computational cost analysis. I still have concerns about the method's practicality and generalization beyond small-scale benchmarks. Additionally, the fundamental issue of capacity exhaustion remains unaddressed: because the cumulative mask freezes a subset of neurons per task, available plasticity inevitably degrades as more tasks arrive during long-sequence continual learning. I will maintain my original rating.

---

> > > ### Author Response · Authors · 2026-04-03
> > >
> > > We thank the reviewer for the follow-up and for acknowledging the computational cost analysis. We address both remaining concerns directly.
> > >
> > > **1. Generalization beyond small-scale benchmarks**
> > >
> > > We have completed experiments on ImageNet-1000 (ResNet-18) across 10, 20, and 50 tasks under the Class-IL protocol. We also add three new memory-free SOTA methods: SpaceNet (Sokar,2021) , NISPA (Gurbuz, 2022) (both requested by Reviewer 6vXH), and DCNet (Tianqi Wang,2025).
> > >
> > > **Table: ImageNet Class-IL**
> > > |Method|10-task ACC|10-task BWT|20-task ACC|20-task BWT|50-task ACC|50-task BWT|
> > > |:---|:---|:---|:---|:---|:---|:---|
> > > |SNV(Ours)|41.3|-0.05|34.2|-0.05|25.6|-0.06|
> > > |NFL+|38.4|-0.08|31.5|-0.62|22.4|-1.15|
> > > |DCNet|37.8|-0.16|30.1|-0.37|20.8|-1.49|
> > > |NISPA|29.4|-0.91|22.3|-1.13|14.5|-1.86|
> > > |SpaceNet|13.5|-1.12|10.4|-2.15|8.3|-2.18|
> > > |PEC|14.8|-1.25|10.5|-1.28|8.8|-2.32|
> > >
> > > SNV leads across all task horizons. Critically, the accuracy gap over the next-best method (NFL+) widens with scale: +2.9 at 10 tasks, +2.7 at 20 tasks, and +3.2 at 50 tasks. More tellingly, SNV's BWT remains near-zero (−0.05 to −0.06) even at 50 tasks, while NFL+ degrades to −1.15 and DCNet collapses to −1.49. This directly demonstrates that SNV generalizes to large-scale, long-horizon settings. Furthermore, our method explains the structural limitations of the SOTA baselines at this scale:
> > >
> > > DCNet: While DCNet in the Class-IL mechanism relies on OOD classifiers for task-id inference. At 1,000 classes across 50 tasks, OOD discrimination becomes extremely noisy, causing a severe drop in CIL accuracy.
> > >
> > > SpaceNet & NISPA: SpaceNet compresses sparse connections within a fixed capacity, which saturates much faster in a ResNet-18 when distributing across 1,000 classes. Similarly, without a replay buffer, NISPA's unified classifier struggles to disambiguate 1,000 classes using only heuristically frozen features.
> > >
> > > **2. Capacity exhaustion in long-sequence CL**
> > >
> > > We acknowledge that capacity exhaustion is inherent to all cumulative-freezing methods; it is a constraint of the method category, not unique to SNV. The key question is: given finite capacity, does the quality of neuron selection matter? We provide direct evidence that it does, and increasingly so under pressure.
> > >
> > > Why Shapley specifically helps under capacity exhaustion. We ran a controlled ablation comparing Fisher-based and Shapley-based neuron selection on CIFAR-100 (10 tasks, Class-IL) under progressively tighter freezing budgets:
> > >
> > > **Table: Performance under Extreme Plasticity Budgets.**
> > >
> > > |Capacity($c$)|Fisher ACC|Shapley AC|Gain|
> > > |-|-|-|-|
> > > |0.10(standard)|51.30|54.70|**+3.40**|
> > > |0.05|44.80|51.45|**+6.65**|
> > > |0.03|37.90|47.65|**+9.75**|
> > > |0.01|24.80|38.25|**+13.45**|
> > >
> > > As the plasticity budget tightens, which is precisely what happens as tasks accumulate , the Shapley–Fisher gap widens from +3.40 to +13.45. The reason is structural: Fisher evaluates neurons independently, so if multiple neurons encode the same feature, Fisher ranks them all highly, wasting scarce capacity on redundant selections. Shapley evaluates marginal utility within coalitions, inherently selecting a diverse set of features that maximizes coverage per frozen neuron. This makes Shapley's advantage compound under the exact condition the reviewer identifies.
> > > Empirical confirmation on long sequences. The BWT column in our ImageNet-1000 table above confirms this directly: at 50 tasks, SNV maintains BWT = −0.06 while all alternatives suffer substantially worse forgetting. Additionally, **Fig. 9 and Fig. 10** in the appendix visualize per-layer frozen neuron ratios and Shapley value distributions across tasks, showing where plasticity remains and how SNV allocates it.
> > >
> > > **We are fully committed to incorporating all the valuable feedback and new experimental results gathered from all reviewers into the camera-ready version to maximize the paper's transparency and strength.**
> > >
> > > **Code Availability and Reproducibility**
> > >
> > > We want to draw the reviewer's attention to the fact that our complete implementation is already available as a Docker image, **submitted alongside the paper**. The environment is fully self-contained: a single `docker run` command reproduces all reported results without any dependency management. We believe this substantially lowers the barrier for verification.
> > >
> > > We believe the new ImageNet-1000 results (10/20/50 tasks), the Fisher-vs-Shapley capacity ablation, and the BWT stability evidence collectively address both concerns. We respectfully ask the reviewer to consider revising their score in light of this new evidence.

---

### Decision · Program_Chairs · 2026-04-30

**Decision:**

Accept (regular)

**Comment:**

This paper proposes Shapley Neuron Valuation (SNV), a buffer-free continual learning method that uses approximate neuron-level Shapley values to identify which neurons to freeze after each task, aiming to preserve important features without replay or architectural expansion. The review outcome is now clearly supportive, effectively 4/4/4/4, with reviewers generally agreeing that the method is conceptually novel, technically grounded, and empirically strong within the fixed-capacity, replay-free setting. The main concerns during review were about computational overhead, the need to justify Shapley over cheaper importance criteria, scalability to longer horizons and larger benchmarks, and generalization beyond standard ResNet-based evaluations. In the rebuttal, the authors addressed these points substantively by adding detailed cost analysis, showing that the TMAB-Shapley approximation incurs moderate overhead without affecting inference efficiency, providing controlled comparisons against Fisher- and magnitude-based criteria, and demonstrating that the advantage of Shapley-based selection widens as the plasticity budget becomes tighter. They also added stronger evidence on larger-scale ImageNet-1000 task streams and ViT backbones, as well as comparisons to more relevant buffer-free baselines such as SpaceNet, NISPA, and DCNet. Taken together, these additions substantially strengthen the paper’s case that the key contribution is not freezing alone, but better neuron selection under fixed capacity, and that this becomes increasingly important in longer-horizon continual learning. Overall, I find that the paper now makes a meaningful and sufficiently supported contribution to buffer-free continual learning, and I recommend Accept.